# Lignans as Pharmacological Agents in Disorders Related to Oxidative Stress and Inflammation: Chemical Synthesis Approaches and Biological Activities

**DOI:** 10.3390/ijms23116031

**Published:** 2022-05-27

**Authors:** Dmitry I. Osmakov, Aleksandr P. Kalinovskii, Olga A. Belozerova, Yaroslav A. Andreev, Sergey A. Kozlov

**Affiliations:** 1Shemyakin-Ovchinnikov Institute of Bioorganic Chemistry, Russian Academy of Sciences, ul. Miklukho-Maklaya 16/10, 117997 Moscow, Russia; osmadim@gmail.com (D.I.O.); kalinovskii.ap@gmail.com (A.P.K.); o.belozyorova@gmail.com (O.A.B.); aya@ibch.ru (Y.A.A.); 2Institute of Molecular Medicine, Sechenov First Moscow State Medical University, 119991 Moscow, Russia

**Keywords:** lignan, antioxidant, oxidative stress, anti-inflammatory, inflammation, chemical synthesis

## Abstract

Plant lignans exhibit a wide range of biological activities, which makes them the research objects of potential use as therapeutic agents. They provide diverse naturally-occurring pharmacophores and are available for production by chemical synthesis. A large amount of accumulated data indicates that lignans of different structural groups are apt to demonstrate both anti-inflammatory and antioxidant effects, in many cases, simultaneously. In this review, we summarize the comprehensive knowledge about lignan use as a bioactive agent in disorders associated with oxidative stress and inflammation, pharmacological effects in vitro and in vivo, molecular mechanisms underlying these effects, and chemical synthesis approaches. This article provides an up-to-date overview of the current data in this area, available in PubMed, Scopus, and Web of Science databases, screened from 2000 to 2022.

## 1. Introduction

Through their vital activity, plants produce a wide range of pharmacologically active natural compounds. Phenylpropane (C_6_C_3_) units, provided by precursor phenylalanine and tyrosine, are found in many natural compounds, including lignans. Lignans are dimer compounds originating from cinnamic acid and its derivatives that also give rise to lignin, pre-eminent polymer component of the plant cell wall. The term “lignans” is often restricted to molecules in which two phenylpropane units are coupled at the central carbon of the side-chain (β-β′-coupling) while compounds with alternative coupling are referred to as neolignans [1]. The focus of this review is the group of lignans. Based on the patterns of cyclization and oxygen incorporation, lignans can be classified into eight subgroups: arylnaphthalenes, aryltetralins, dibenzocyclooctadienes, dibenzylbutanes, dibenzylbutyrolactones, dibenzylbutyrolactols, furans, and furofurans (Figure 1). Along with the diversity of structure, lignans exert a broad spectrum of biological activities, e.g., antitumor, antiviral, hepatoprotective, immunosuppressive, anti-platelet, and cardiovascular effects [2]. Additionally, some lignans can produce strong antioxidant and anti-inflammatory effects.

Oxidative stress is an imbalance of the oxidants/antioxidants tilting toward an oxidative status, which is characterized by a higher level of reactive oxygen species (ROS) and reactive nitrogen species (RNS) than in the normal physiological state. It could be triggered by heavy metals, xenobiotics, free radicals, drugs, and ionizing radiation. Exposure to these toxicants and oxidants impairs cellular components (e.g., lipids, proteins, and nucleic acids) and initiates the pathogenesis of diabetes mellitus, cancer, neurodegenerative, cardiovascular, lung diseases, etc. [3].

Inflammation is an adaptive response induced by pathogens, tissue damage or ingestion of allergens or pollutants that includes activation of innate and adaptive immunity. This process is coordinated by a complex regulatory network of factors that fall into four functional categories: inducers, sensors, mediators, and effectors. Endogenous and exogenous inducers activate specialized sensors, e.g., toll-like receptors, inflammasome, IgE, TRP and ASIC channels, etc. They, in turn, elicit the production of specific sets of mediators (vasoactive amines, vasoactive peptides, fragments of complement components, eicosanoids, cytokines, chemokines, and proteolytic enzymes) that alter the functionality of tissues and organs (downstream effectors). Irrespective of injury or infection, long-term stress and malfunction of tissues can also induce chronic low-level inflammation that is common to such disorders as obesity, type 2 diabetes, atherosclerosis, asthma, cancer, autoimmune and neurodegenerative diseases [4]. The pathological effect of simultaneously developing oxidative stress and inflammation is enormous and requires pharmacological measures for control.

## 2. Molecular Mechanisms of Inflammation and Oxidative Stress

Signals of inflammation can be divided into damage-associated molecular patterns (DAMP), which are cell-derived and initiate immunity in response to trauma and tissue damage, and pathogen-associated molecular patterns (PAMP), which are derived from microorganisms [5,6]. They stimulate protective reactions via the interaction with pattern recognition receptors with the subsequent formation of inflammasomes. Activation of the inflammasome requires two events: priming and activation. For example, the priming can include recognition of a bacterial lipopolysaccharide (LPS) by toll-like receptor 4 (TLR4) which leads to the activation of a cascade of reactions resulting in the translocation of nuclear factor kappa-light-chain-enhancer of activated B (NF-κB) into the cell nucleus (Figure 2). Activation of NF-κB is the general event in the inflammatory reaction of a cell and includes two main signaling pathways: canonical and non-canonical (or alternative). The main mechanism of the canonical activation of NF-κB is the induced degradation of the NF-κB-IκB complex triggered by site-specific phosphorylation by IκB kinase (IKK) [7]. The inhibitory subunit IκBα then leaves the complex and, via ubiquitinylation, is degraded in the proteasome [8]. Non-canonical activation of NF-κB does not involve IκBα degradation, but depends on the processing of NF-κB2 precursor protein, called p100, by NF-κB-inducing kinase (NIK), which activates and functionally interacts with IKK. The processing of p100 with the degradation of its C-terminal IκB-like structure leads to the formation of a mature NF-κB2 p52 and the nuclear translocation of this non-canonical NF-κB complex into the nucleus [9]. Canonical or alternatively activated NF-κB in the nucleus promotes the transcription of NF-κB-dependent genes, such as the NLR family pyrin domain containing 3 (NLRP3), pro-Il-1ß and pro-Il-18, which are necessary for the activation of the inflammasome. Mitochondrial damage, as well as plasma membrane damage and K+ efflux, are also considered upstream mechanisms that regulate inflammasome activation [10]. Finally, activated inflammasome produces inflammatory mediators for excretion, which includes the maturation of pro-inflammatory cytokines such as interleukin-1-beta (IL-1β), IL-6, and tumor necrosis factor-alpha (TNF-α) by caspase-1.

Nuclear factor erythroid-related factor 2 (Nrf2) is a redox-sensitive transcription factor that plays an essential role in the protection against oxidative stress and electrophilic injury by regulating a battery of cytoprotective genes. Under basal conditions, Nrf2 binds to its repressor Kelch-like ECH-associated protein 1 (Keap1) and is maintained at a low level in the cytosol through Keap1-mediated ubiquitinylation and 26S proteasome-mediated degradation. The activation of the Nrf2-mediated defensive response is an effective means of counteracting exogenous oxidative insults. Electrophilic and oxidative stressors, such as ROS, can activate Nrf2 promoting its dissociation from Keap1 or phosphorylation by several kinases, including advanced protein kinase B (Akt), extracellular signal-regulated kinase (ERK), p38 mitogen-activated protein kinase (p38 MAPK), and protein kinase C (PKC). The activated Nrf2 is guided into the nucleus, forms a heterodimer with a small musculo-aponeurotic fibrosarcoma (Maf) protein, binds to specific DNA sequences called the antioxidant responsive element (ARE) consensus and subsequently initiates the transcription of downstream cytoprotective genes. These ARE-containing genes include various redox-balancing proteins and phase II enzymes, such as heme oxygenase-1 (HO-1) and NAD(P)H:quinone oxidoreductase 1 (NQO1), γ-glutamyl cysteine synthetase (γ-GCS) and glutamate-cysteine ligase (GCL), thioredoxin (Trx), thioredoxin reductase (TrxR) and peroxiredoxin (Prx) as well as glutathione (GSH)-utilizing enzymes, such as superoxide dismutase (SOD), glutathione peroxidase (GPx) and catalase (CAT), glutathione S-transferase (GST), glutathione reductase (GR). These proteins maintain the cellular redox capacity, eliminate ROS, promote excretion of toxicants, and ensure cytoprotection [11,12,13].

The inflammation and ROS protection cascades regulate each other in several ways (Figure 2). For example, the Nrf2 transcription factor has been shown to negatively control the NF-κB signaling pathway through various mechanisms. First, Nrf2 inhibits the oxidative stress-mediated activation of NF-κB by reducing intracellular ROS levels [14]. The activation of Nrf2 has also been reported to inhibit the LPS-induced production of pro-inflammatory cytokines, including IL-6 and IL-1ß, through an ROS-independent mechanism. This was due to the negative regulation of NF-IκB-mediated transcription of pro-inflammatory cytokine genes and genes involved in the inflammasome assembly, such as NLRP3 and caspase 1 [15]. In addition, Nrf2 prevents IκB-α from degradation, thereby inhibiting the nuclear translocation of NF-κB [16]. NF-κB, in turn, prevents transcriptional co-activators from binding to Nrf2 and subsequent ARE transcription [17]. It was also found that GSH suppresses the activation of p38, as well as the expression of cyclooxygenase-2 (COX-2) in peritoneal macrophages of rats exposed to LPS stimulation [18], which shows that the content of GSH can strongly affect the activity and function of molecular and cellular mediators of inflammatory processes.

The mitochondrial status also plays an important role in the regulation of both inflammatory and oxidative stress processes. The degradation of mitochondria through the Bax/Bak pathway leads to an increase in intracellular ROS, oxidized lipids and mtDNA, which provokes the formation of the inflammasome and NF-κB-mediated inflammatory response. Mitochondrial biogenesis, as well as glucose transport and fatty acid oxidation, is improved by the transcription factor PGC-1α. The increase in both AMP/ATP and NAD+/NADH ratio triggers PGC-1α activation through its AMPK-mediated phosphorylation and SIRT1-mediated deacetylation [19,20]. PGC-1α can also activate Nrf2 by inhibiting the GSK3ß regulator, which prevents the translocation of Nrf2 into the nucleus by phosphorylation. Under oxidative stress, GSK3β is inactivated by p38, which is positively regulated by PGC-1α, and, as a consequence, the antioxidant defense gets activated by Nrf2 [21].

p53 is another protein whose activity depends on the levels of ROS/RNS in oxidative stress conditions. p53 acts as a metabolic regulator or even an apoptosis inducer [22]. AMPK and p38 MAPK can phosphorylate p53 and induce its interaction with PGC-1α to enhance Nrf2 response and reduce the effects of ROS/RNS [23].

## 3. Lignans with Antioxidant and Anti-Inflammatory Action

As shown in Figure 2, the response to ROS and RNS is unidirectional with the anti-inflammatory response. The antioxidant and anti-inflammatory effects of lignans measured in models have been summarized by us in separate tables for a better understanding. The antioxidant effects measured in vitro and ex vivo/in vivo are presented in Table 1 and Table 2, respectively. Data on anti-inflammatory action obtained in experiments in vitro and ex vivo/in vivo are presented in Table 3 and Table 4, respectively.

**Table 1 ijms-23-06031-t001:** Antioxidant activity of lignans in vitro.

Lignan	Source	Model/Assay	Target	Concentration	Ref.
**Aryltetralinstructure group**					
(−)-Isoguaiacin	*Machilusthunbergii* Sieb, et Zucc.	CCl_4_-induced hepatotoxicity	↓GPT level	50–100 μM	[24]
			↓MDA content, ↑GSH/GSSG level, ↑SOD1, ↑CAT	50 μM	
(+)-Isolariciresinol	Riesling wine	TEAC assay	radical scavenging capacity	2.5 mmol Trolox/mmol	[25]
	*Ephedra viridis*	DCFH assay in HL-60 cells	↓iROS level	IC_50_ 21 μg/mL	[26]
	*Euterpe oleracea* Mart.	HO assay	HO^•^ scavenging	IC_50_ 0.68 ± 0.02 μg/mL	[27]
		DPPH assay	DPPH radical scavenging	IC_50_ 37.4 ± 0.9 μg/mL	
(±)-Isolariciresinol	Synthetic	DPPH assay	DPPH radical scavenging	IC_50_ 53.0 μM	[28]
(−)-Isolariciresinol 5-methoxy-9-β-D-xylopyranoside	*Saracaasoca* (Roxb.) De Wilde	DPPH assay	DPPH radical scavenging	IC_50_ 44 μM	[29]
(+)-Isolariciresinol 3a-O-β-D-glucopyranoside	*Carissa spinarum* Linn.	DPPH assay	DPPH radical scavenging	IC_50_ 45.7 ± 1.5 μM	[30]
		FRAP assay	ferric-reducing potentiality	33 mmol Fe^2+^/g	
		H_2_O_2_-induced L02 cells cytotoxicity	↓iROS level	5 μM	
Isolariciresinol-9’-O-α-L-arabinofuranoside	*Pinus massoniana* Lamb.	H_2_O_2_-induced HUVECs cytotoxicity	↑PI3K, ↑p-Akt, ↑p-Bad, ↓Bax	31.3–125 μg/mL	[31]
Lyoniresinol	*Berberis vulgaris* Linn.	HO assay	HO^•^ scavenging	IC_50_ 1.4 ± 0.12 μg/mL	[32]
	*Viscum album* Linn.	ABTS assay	ABTS radical scavenging	10–100 μM	[33]
		DPPH assay	DPPH radical scavenging		
		Glu-treated HT22 cells	↓iROS level	25 μM	
(+)-Lyoniresinol-3α-O-β-glucopyranoside	*Strychnosvanprukii*	DPPH assay	DPPH radical scavenging	IC_50_ 31.2 μM	[34]
Sauchinone	Synthetic	AngII-induced mesangial cells	↓iROS level	1 μM	[35]
**Dibenzocyclooctadiene structure group**					
Gomisin A	Synthetic	ZnPP/high glucose-injured MC3T3 E1 cells	↓iROS level, ↑SOD, ↑HO-1	1–10 μM	[36]
Gomisin N	Synthetic	HeLa cells	↑iROS level	100 μM	[37]
		ethanol-treated HepG2 cells	↓iROS level, ↑GSH/GSSG level, ↑CAT, ↑SOD, ↑GPx ↑SIRT1/AMPK, ↓CYP2E1	50–100 μM	[38]
Schisandrin A	*Schisandra chinensis* Baill.	CCl_4_-treated HepG2 cells	↓TBARS level, ↓iROS level	50 μM	[39]
	Synthetic	LPS-stimulated RAW 264.7 macrophages	↓iROS level ↓Keap1, ↑Nrf2, ↑HO-1	200 μM	[40]
		H_2_O_2_-induced C2C12 cell cytotoxicity	↓iROS level, ↑AMPK, ↑Bcl-2/Bax	200 μM	[41]
		DON-induced cytotoxicity in HT-29 cells	↓iROS level, ↓TBARS level, ↓CAT, ↓SOD, ↓GPx, ↑Nrf2, ↑HO-1, ↑GST, ↑GSH/GSSG level	2.5–10 μM	[42]
		RANKL-induced osteoclast differentiation model	↓iROS level, ↑Nrf2, ↑HO-1, ↑CAT ↓TRAF6, ↓Nox1	50–200 μM	[43]
Schisandrin B	*Schisandra chinensis* (Turcz.) Baill.	CCl_4_-treated HepG2 cells	↓TBARS level	50 μM	[39]
			↓iROS level	10–50 μM	
			↑CYP3A4 expression and activity	50 μM	
	Synthetic	PQ-induced PC12 cells cytotoxicity	↓iROS level, ↑GSH/GSSG level	15 μM	[44]
		solar-irradiated BJ human fibroblast	↓iROS level, ↓MMP, ↑GSH/GSSG level	25–75 μM	[45]
		intact lymphocytes	↑iROS level, ↓GSH/GSSG level, ↑Nrf2, ↑HO-1, ↑TR, ↑GCLC	25–50 μM	[46]
		H_2_O_2_-induced PC12 cells cytotoxicity	↓iROS level, ↓MDA content, ↑SOD	2.5–10 μM	[47]
			↑Bcl-2/Bax, ↑p-Akt/Akt	10 μM	
		CsA-induced cytotoxicity in HK-2 cells	↓iROS level, ↑GSH/GSSG level, ↑Nrf2, ↑HO-1, ↑NQO1, ↑GCLM ↑Bcl-2/Bax	2.5–10 μM	[48]
		tBHP-induced HaCaT cell injury	↓iROS level, ↑Nrf2, ↑HO-1, ↑SOD, ↑GPx, ↑CAT, ↑p-AMPK, ↑p-Akt, ↑p-Erk1/2, ↑p-JNK, ↑p-p38	2.5–10 μM	[49]
		H/R-induced H9c2 cell injury	↓iROS level, ↑SOD, ↑GPx, ↑Nrf2, ↑NQO-1, ↑HO-1, ↓Keap1, ↑AMPK	20 μM	[50,51]
Schisandrin C	Synthetic	solar-irradiated BJ human fibroblast	↓iROS level, ↓MMP, ↑GSH/GSSG level	25–75 μM	[45]
		LPS-stimulated HDPCs	↓iROS level, ↑SOD ↑HO-1, ↑PGC-1α, ↑Nrf2, ↑p-Akt	10–20 μM	[52]
Schisantherin A	*Schisandra chinensis* (Turcz.) Baill.	CCl_4_-treated HepG2 cells	↓TBARS level	50 μM	[39]
			↓iROS level	2–50 μM	
	*Schisandra sphenanthera*	H/R-induced HK-2 cells	↓iROS level, ↑SOD, ↑MDA content ↑Bcl2/Bax, ↑PI3K/AKT	5–20 μM	[53]
		LPS-stimulated BV-2 microglial cells	↓iROS level, ↑HO-1, ↑NQO-1	2.5–50 μM	[54]
			↑Nrf2	50 μM	
	Synthetic	LPS-stimulated NRK-52E cells	↑γGCS, ↑Nrf2	25–50 μM	[55]
**Dibenzylbutane structure group**					
(–)-Carinol	*Carissa spinarum* Linn.	DPPH assay	DPPH radical scavenging	IC_50_ 20.2 μM	[56]
	Synthetic	DPPH assay	DPPH radical scavenging	IC_50_ 4.4 μg/mL	[57]
		XOD assay	↓xanthine oxidase enzyme	IC_50_ 219.4 μg/mL	
Meso-dihydroguaiaretic acid	*Machilusthunbergii* Sieb, et Zucc.	CCl_4_-induced hepatotoxicity	↓GPT level	10–100 μM	[24]
			↓MDA content, ↑GSH/GSSG level, ↑SOD1, ↑CAT	50 μM	
	*Machilusphilippinensis* Merr.	fMLF-activated human neutrophils	↓O_2_^•–^ level	IC_50_ 0.78 ± 0.17 μM	[58]
			↓iROS level	IC_50_ 0.79 ± 0.26 μM	
			↓p-ERK, ↓p-JNK, ↓p-Akt	10 μM	
		MMK-1-activated human neutrophils	↓O_2_^•–^ level	IC_50_ 1.17 ± 0.64 μM	
		PMA-activated human neutrophils	↓iROS level	IC_50_ 3.57 ± 3.93 μM	
		ABTS assay	ABTS radical scavenging	1–10 μM	
		DPPH assay	DPPH radical scavenging		
		ORAC assay	ROS scavenging		
		XOD assay	superoxide anion scavenging		
Nordihydroguaiaretic acid	*Larrea tridentate*	DCFH assay in HL-60 cells	↓iROS level	IC_50_ 0.7 μg/mL	[59]
	Synthetic	FL5.12 cells	↑p-ERK1/2, ↑p-JNK, ↑p-p38	20 μM	[60]
		HOCl assay	hypochlorous acid scavenging	IC_50_ 622 ± 42 μM	[61]
		O_2_^•–^ assay	superoxide anion scavenging	IC_50_ 15 ± 1 μM	
		OH assay	OH radical scavenging	IC_50_ 0.15 ± 0.02 μM	
		^1^O_2_ assay	singlet oxygen scavenging	IC_50_ 151 ± 20 μM	
		ONOO assay	ONOO anion scavenging	IC_50_ 4 ± 0.94 μM	
		H_2_O_2_/3-NP-induced CGNs neurotoxicity	↑Nrf2, ↑HO-1	20 μM	[62]
		OH assay	OH radical scavenging	10 μM	[63]
		TPA-treated mouse model	↑GPx, ↑GR, ↑GST, ↑GSH/GSSG level, ↑SOD, ↑CAT	15–25 μM	[64]
		H_2_O_2_-induced LLC-PK1/MEFs cells cytotoxicity	↓iROS level, ↑Nrf2, ↑HO-1 ↑p-Akt, ↑p-ERK1/2, ↑p-p38, ↑p-JNK, ↑p-GSK-3	15 μM	[65]
		IAA/H_2_O_2_-induced cytotoxicity in MN and THP-1 cells	↓iROS level, ↑GSH/GSSG level ↑CD33	20 μM	[66]
		Daoy cells	↑GSH/GSSG level	75 μM	[67]
(–)-Secoisolariciresinol	*Taxus yunnanensis*	DPPH assay	DPPH radical scavenging	IC_50_ 28.9 μM	[68]
	*Araucaria angustifolia*	rat liver microsomes	↓lipid peroxidation activity	IC_50_ 0.1–0.15 μM	[69]
		O_2_^•–^ assay	superoxide anion scavenging	IC_50_ 4.8 nM	
		ROO assay	peroxyl radicals scavenging	SF 3.1–4.0 mole/mole	
	*Piceaabies*	DPPH assay	DPPH radical scavenging	IC_50_ 9.0 ± 1.0 μM	[70]
	*Linum usitatissimum* Linn.	L-α-phosphatidylcholine liposome/pBR322 plasmid DNA	AAPH radical scavenging	50–100 μM	[71]
			DPPH radical scavenging	25–200 μM	
	*Carissa spinarum* Linn.	DPPH assay	DPPH radical scavenging	IC_50_ 26.2 μM	[56]
Secoisolariciresinol-7-hydroxyl	*Piceaabies*	DPPH assay	DPPH radical scavenging	IC_50_ 12.7 ± 1.5 μM	[70]
Secoisolariciresinol diglucoside	*Linum usitatissimum* Linn.	L-α-phosphatidylcholine liposome/pBR322 plasmid DNA	AAPH radical scavenging	10–100 μM	[71]
			DPPH radical scavenging	25–200 μM	
	Synthetic	DPPH assay	DPPH radical scavenging	IC_50_ 78.9 ± 0.29 μg/mL	[72]
	*Linum usitatissimum* Linn.	iron treated H9c2 cells	↓iROS level, ↑SOD, ↑Bcl-2/Bax ↓MMP-2, ↓MMP-9, ↓FOXO3a, ↓p70S6K1, ↑AMPK	500 μM	[73]
	Synthetic (LGM2605)	asbestos-exposed MFs	↓iROS level, ↓MDA content, ↓8-isoP, ↑Nrf2, ↑NQO-1, ↑HO-1, ↑GST, ↑TR, ↓nitrate/nitrite ratio	50–100 μM	[74,75]
		LPS-stimulated AC16 cells	↓iROS level	50 μM	[76]
**Dibenzylbutyrolactone structure group**					
Arctigenin	Synthetic	glutamate-treated rat cortical cells	↓iROS level	IC_50_ 33.2 μM	[77]
		LPS-treated Raw264.7 cells	↓iROS level	5–50 μM	[78]
	*Arctium lappa* Linn.	glucose-starved A549 cells	↓iROS level	10 μM	[79]
		H_2_O_2_-treated L6 cells	↑Nrf2, ↑SOD, ↑GR, ↑GPx, ↑Trx1, ↑UCP2, ↑p-AMPK, ↑p-p53, ↑p21, ↑PGC-1α, ↑PPARα	1–20 μM	[80]
	Synthetic	MDA-MB-231 cells	↑iROS level, ↓GSH/GSSG level, ↑Nox, ↑p-p38, ↑p-ATF-2, ↓Bcl-2	5 μM	[81]
		H_2_O_2_-treated astrocytes	↓iROS level	10–20 μM	[82]
		intact astrocytes	↑HO-1, ↑Nrf2, ↑c-Jun, ↑p-Akt		
		TGF-β1-induced HK-2 cells	↓iROS level, ↓Nox ↓p-Akt, ↓p-ERK1/2, ↓p-IκBα	0.5–1 μM	[83]
	*Arctium lappa* Linn.	DPPH assay	DPPH radical scavenging	IC_50_ 31.47 ± 2.33 μM	[84]
		H2DCF-DA assay	↓iROS level	10–100 μM	
	Synthetic	OA-treated WRL68 hepatocytes	↓MDA content ↑p-PI3K, ↑p-Akt, ↑p-AMPK	50 μM	[85]
		Hep G2 cells	↑iROS level, ↓GSH/GSSG level	5–100 μM	[86]
			↑p-p38, ↑p-JNK	20 μM	
		OGD-injured H9c2 cardiomyocytes	↓iROS level, ↓MDA content, ↑SOD ↑AMPK/SIRT1	50–200 μM	[87]
		silica-injured RAW 264.7 macrophages	↓iROS level	1 μM	[88]
Hinokinin	Synthetic	antioxidant assay	inhibition of H_2_O_2_ produced by Trypanosoma cruzi mitochondria	IC_50_ 17.84 μM	[89]
Matairesinol	*Cedrus deodara*	DPPH assay	DPPH radical scavenging	IC_50_ 33.24 ± 0.47 μM	[90]
	*Piceaabies*	rat liver microsomes	↓lipid peroxidation activity	IC_50_ 0.28 μM	[69]
		O_2_^•–^ assay	superoxide anion scavenging	IC_50_ 40 nM	
		ROO assay	peroxyl radicals scavenging	SF 1.0 mole/mole	
		DPPH assay	DPPH radical scavenging	IC_50_ 14.0 ± 0.0 μM	[70]
	*Arctium lappa*	DPPH assay	DPPH radical scavenging	IC_50_ 14.95 ± 0.38 μM	[84]
		H2DCF-DA assay	↓iROS level	100 μM	
	Synthetic	DPPH assay	DPPH radical scavenging	20 μM	[91]
		O_2_^•–^ assay	superoxide anion scavenging		
		hypoxia-induced HeLa cells	↓miROS levels ↓HIF-1α, ↓VEGF	10–50 μM	[92]
		LPS-stimulated NSC-34 neurons and BV2 microglia	↓MDA content, ↑SOD, ↑CAT, ↑GPx, ↑Nrf2, ↑HO-1, ↑AMPK	5–20 μM	[93]
Matairesinol-7′-hydroxyl	*Piceaabies*	rat liver microsomes	↓lipid peroxidation activity	IC_50_ 0.15–0.18 μM	[69]
		O_2_^•–^ assay	superoxide anion scavenging	IC_50_ 217 nM	
		ROO assay	peroxyl radicals scavenging	SF 2.1–2.7 mole/mole	
		DPPH assay	DPPH radical scavenging	IC_50_ 15.7 ± 0.6 μM	[70]
		DPPH assay	DPPH radical scavenging	IC_50_ 20.0 ± 0.1 μM	[94]
(+)-Nortrachelogenin	*Wikstroemia indica*	DPPH assay	DPPH radical scavenging	IC_50_ 90.1 μM	[95]
(–)-Nortrachelogenin	*Cedrus deodara*	DPPH assay	DPPH radical scavenging	IC_50_ 36.79 ± 1.69 μM	[90]
	*Pinus contorta*	rat liver microsomes	↓lipid peroxidation activity	IC_50_ 0.14–0.19 μM	[69]
		O_2_^•–^ assay	superoxide anion scavenging	IC_50_ 1.4 nM	
		ROO assay	peroxyl radicals scavenging	SF 2.0–2.2 mole/mole	
	*Piceaabies*	DPPH assay	DPPH radical scavenging	IC_50_ 17.7 ± 1.5 μM	[70]
	*Carissa carandas* Linn.	DPPH assay	DPPH radical scavenging	IC_50_ 30.2 μM	[96]
	*Carissa spinarum* Linn.	DPPH assay	DPPH radical scavenging	IC_50_ 35.8 μM	[56]
	*Galactites elegans*	DPPH assay	DPPH radical scavenging	IC_50_ 38.6 ± 2.7 μM	[97]
		BHP-treated Jurkat cells	peroxyl radicals scavenging	50 μM	
**Furanoid structure group**					
(+)-Lariciresinol	*Abies balsamea*	rat liver microsomes	↓lipid peroxidation activity	IC_50_ 0.17–0.35 μM	[69]
		O_2_^•–^ assay	superoxide anion scavenging	IC_50_ 35 nM	
		ROO assay	peroxyl radicals scavenging	SF 1.0–2.6 mole/mole	
	*Hemerocallis fulva*	LUVs assay	↓lipid peroxidation activity	50 μg/mL	[98]
	*Piceaabies*	DPPH assay	DPPH radical scavenging	IC_50_ 10.7 ± 1.2 μM	[70]
	*Ephedra viridis*	DCFH assay in HL-60 cells	↓iROS level	IC_50_ 17.7 μg/mL	[26]
	*Euterpe oleracea* Mart.	HO assay	HO^•^ scavenging	IC_50_ 0.70 ± 0.13 μg/mL	[27]
		DPPH assay	DPPH radical scavenging	IC_50_ 22.4 ± 3.0 μg/mL	
	*Rubia philippinensis*	ABTS assay	ABTS radical scavenging	12.5–50 μM	[99]
		DPPH assay	DPPH radical scavenging		
		HO assay	HO^•^ scavenging	1.5–6 μM	
		ORAC assay	↓ROO^•^-induced oxidation		
		CUPRAC assay	cupric-reducing potentiality	6.25–50 μM	
		FRAP assay	ferric-reducing potentiality		
		AAPH-treated RAW 264.7 cells	↓iROS level	12.5–50 μM	
		RAW 264.7 cells	↑Nrf2, ↑SOD1, ↑CAT, ↑GPx, ↑HO-1, ↑NQO1, ↑GCLc, ↑GCLm ↑p-p38, ↑p-ERK1/2		
Nectandrin B	*Myristica fragrans*	DPPH assay	DPPH radical scavenging	5–50 μg/mL	[100]
		old HDFs	↓p-AMPK, ↑p-PI3K, ↑p-Akt, ↓p-ERK1/2, ↓p-p38	10–20 μg/mL	
		H_2_O_2_/palmitic acid-treated old HDFs	↓iROS level, ↑SOD1,2		
(−)-Olivil	*Carissa spinarum* Linn.	DPPH assay	DPPH radical scavenging	IC_50_ 18.1 μM	[56]
Taxiresinol	*Taxus yunnanensis*	DPPH assay	DPPH radical scavenging	IC_50_ 18.4 μM	[68]
**Furofuranoid structure group**					
4-ketopinoresinol	*Coixlachryma-jobi* Linn. var. *ma-yuen* Stapf	DPPH assay	DPPH radical scavenging	IC_50_ 52.7 ± 4.6 μg/mL	[101]
		H_2_O_2_-induced HSC-3 cell cytotoxicity	↓iROS level	12.5–50 μM	[102]
			↑GSH/GSSG level	50 μM	
			↑Nrf2	6.25–50 μM	
			↑HO-1, ↑AKR1C1-3, ↑ABCC2, ↑GR, ↑GCLC, ↑GCLM, ↑TR, ↑ABCC5, ↑PI3K/Akt	25 μM	
	*Galactites elegans*	DPPH assay	DPPH radical scavenging	IC_50_ 143.3 ± 13.1 μM	[97]
		BHP-treated Jurkat cells	peroxyl radicals scavenging	50 μM	
Dendranlignan A	*Dendranthema morifolium* (Ramat.)	LPS-induced H9c2 cells	↓iROS level	10 μM	[103]
Isoeucommin A	*Eucommia ulmoides* Oliv.	high-glucose-stimulated HRMCs	↓MDA content, ↑SOD,↑p-GSK-3β, ↑Nrf2, ↑HO-1	62.5–125 μM	[104]
Koreanaside A	*Forsythia koreana*	ORAC assay	↓ROO^•^-induced oxidation	25 μg/mL	[105]
		MOVAS cells	↓VCAM-1		
Pinoresinol	*Forsythia suspensa* (Thunb.)	Cu2+-induced LDL lipid peroxidation	↓lipid peroxidation activity	IC_50_ 1.39 μM	[106]
	*Eucalyptus globulus* Labill	rat liver microsomes	↓lipid peroxidation activity	IC_50_ 7.9 μg/mL	[107]
	*Piceaabies*	DPPH assay	DPPH radical scavenging	IC_50_ 17.7 ± 0.6 μM	[70]
	*Euterpe oleracea* Mart.	DPPH assay	DPPH radical scavenging	IC_50_ 34.7 ± 5.0 μg/mL	[27]
		HO assay	HO^•^ scavenging	IC_50_ 1.8 ± 0.2 μg/mL	
	*Carissa spinarum* Linn.	DPPH assay	DPPH radical scavenging	IC_50_ 43.4 μM	[56]
	*Forsythia koreana*	ORAC assay	↓ROO^•^-induced oxidation	25 μg/mL	[105]
	*Galactites elegans*	DPPH assay	DPPH radical scavenging	IC_50_ 50.8 ± 3.1 μM	[97]
	Cinnamon	intact Beas-2B cells	↑Nrf2, ↑NQO1, ↑γ-GCS	25 μM	[108]
		As(III)-induced Beas-2B cells injury	↓iROS level, ↑GSH/GSSG level		
Pinoresinol diglucoside	*Eucommia ulmoides*	oxLDL-induced HUVECs cytotoxicity	↓iROS level, ↓MDA content, ↑SOD	1 μM	[109]
Sesamin	*Sesamum indicum* Linn.	oxLDL-induced HUVECs cytotoxicity	↓iROS level, ↑SOD1 ↑Bcl-2/Bax level	12.5–100 μM	[110]
	Synthetic	KA-induced PC12 and BV-2 cells	↓iROS level, ↓MDA content	0.1–2 μM	[111]
	*Sesamum indicum* Linn.	dexamethasone-treated osteoblasts	↓iROS level, ↑Bcl-2/Bax, ↑p-Akt	5–20 μM	[112]
		H_2_O_2_-induced Caco-2 cell cytotoxicity	↓iROS, ↑GSH/GSSG level, ↓MDA content, ↑SOD, ↑Nrf2, ↓Keap1, ↑HO-1, ↑NQO1, ↑GCLC, ↑GCLM, ↑GR, ↑p-AKT, ↑p-ERK1/2	20–80 μM	[113]
		H_2_O_2_-induced SH-SY5Y cell cytotoxicity	↓iROS level, ↑SOD2, ↑CAT ↑FoxO3a, ↑SIRT1, ↑SIRT3	1 μM	[114]
Syringaresinol	*Coixlachryma-jobi* Linn. var. *ma-yuen* Stapf	DPPH assay	DPPH radical scavenging	IC_50_ 24.6 ± 3.1 μg/mL	[101]
	*Euterpe oleracea* Mart.	DPPH assay	DPPH radical scavenging	IC_50_ 29.7 ± 2.0 μg/mL	[27]
		HO assay	HO^•^ scavenging	IC_50_ 0.40 ± 0.13 μg/mL	
	*Panax ginseng* C.A. Meyer	H/R-induced H9c2 cells	↓iROS level, ↑MnSOD, ↑CAT, ↑LC3, ↑Bcl-2/Bax, ↓HIF-1, ↑FoxO3a, ↓BNIP3, ↓cCYC, ↑mCYC	25 μM	[115]
	*Sargentodoxa cuneata*	high glucose-injured NRVMs	↑Nrf2, ↑NQO-1, ↑HO-1, ↓Keap1, ↑SOD, ↑Bcl-2/Bax	50–100 μM	[116]

3-NP, 3-nitropropionic acid; 8-isoP, 8-iso prostaglandin F2α; γ-GCS, γ-glutamyl cysteine synthetase; AAPH assay, 2,20-azo-bis(2-amidinopropane) dihydrochloride radical-scavenging method; ABCC, ATP-dependent drug efflux pumps (ATP-binding cassette), subfamily C (CFTR/MRP) members; ABTS assay, 2,2′-azino-bis(3-ethylbenzothiazoline)-6-sulfonic acid radical-scavenging method; AC16 cells, human ventricular cardiomyocyte-derived cell line; AKR1C1-3, aldo-keto reductase 1 subunits C-1-3; AKT, protein kinase B; AMPK, AMP-activated protein kinase; ATF-2, activator protein 1 (AP-1) transcription factor; BNIP3, Bcl-2 interacting protein 3; BV-2 cells, murine microglial cell line; CAT, catalase; CD33, myeloid cell-specific type I transmembrane glycoprotein; CsA, cyclosporine A; CUPRAC assay, cupric-reducing antioxidant capacity method; CYC c/m, cytochrom c in the cytosolic/mitochondrial fraction; CYP, cytochrome P450; DCFH-DA assay, 2′,7′-dichlorofluorescin diacetate radical-scavenging method; DON, trichothecene toxin deoxynivalenol; DPPH assay, 1,1-diphenyl-2-picrylhydrazyl radical-scavenging method; ERK, extracellular signal-regulated kinase; fMLF, N-Formyl-Met-Leu-Phe; Fox, class O forkhead/winged helix transcription factor; FRAP assay, ferric reducing antioxidant power method; GCLC/M, γ-glutamylcysteine synthetase catalytic/modifier subunit; GPT level, serum glutamic pyruvic transaminase content; GPx, glutathione peroxidase; GR, glutathione reductase; GSH/GSSG level, ratio of reduced glutathione with oxidized glutathione; GSK-3, glycogen synthase kinase-3; GST, glutathione-S-thansferase; H2DCF-DA assay, 2,7-dichlorodihydrofluorescein-diacetate; H9c2 cells, embryonic rat heart derived cell line; HDF, human diploid fibroblast; HDPCs, human dental pulp cells; HIF-1, hypoxia induction factor 1; HK-2, human renal tubular epithelial cells; HO-1, heme oxygenase-1; HO assay, hydroxyl radical scavenging method; H/R, hypoxia/reoxygenation; HRMCs, human renal mesangial cells; HUVECs, human umbilical vein endothelial cells; IAA, iodoacetate; IκBα, inhibitor of κB; JNK, c-Jun N-terminal kinase; KA, kainic acid; Keap1, kelch-like ECH-associated protein 1; LC3, microtubule-associated proteins 1A/1B light chain 3B; LPS, lipopolysaccharide; LUVs assay, inhibition of the oxidation of large unilamellar vesicles method; MDA, malondialdehyde; MDA-MB-231, ER-negative breast adenocarcinoma cells; MEFs, mouse embryo fibroblasts; MMK-1, Leu-Glu-Ser-Ile-Phe- Arg-Ser-Leu-Leu-Phe-Arg-Val-Met; MMP, matrix metalloproteinase; MnSOD, manganese superoxide dismutase; MOVAS, mouse vascular smooth muscle cell line; Nox, NADPH oxidase; NQO1, NADPH quinone acceptor oxidoreductase 1; Nrf2, nuclear transcription factor-erythroid 2 related factor; NRVMs, neonatal rat ventricular myocytes; OA, oleic acid; OGD, oxygen glucose deprivation; ORAC, oxygen radical absorbance capacity; ORAC antioxidant assay, inhibition of peroxyl-radical-induced oxidation initiation by thermal decomposition of AAPH; oxLDL, oxidized low-density lipoprotein; p70S6K1, p70S6 Kinase 1; PC12 cells, rat pheochromacytoma cell line; PGC-1α, peroxisome proliferator-activated receptor gamma coactivator 1-alpha; PI3K/AKT pathway, phosphatidylinositol 3-kinase/protein kinase B signaling; PMA, phorbol 12-myristate 13-acetate; POX, peroxidase; PPAR, peroxisome proliferator-activated receptor; PRDx3, peroxiredoxin 3; RANKL, receptor activator of NF-κB ligand; ROO assay, peroxyl radicals scavenging method; iROS, intracellular reactive oxygen species; SF, stoichiometric factor (moles peroxyl radicals scavenged per mole of compound); SH-SY5Y, human neuroblastoma cell line; SIRT1, NAD-dependent deacetylase sirtuin-1; SOD, superoxide dismutase; SOD1, cytoplasmic copper/zinc superoxide dismutase; SOD assay, superoxide radical scavenging method; TBARS, thiobarbituric acid reactive substance; tBHP, tert-butylhydroperoxide; TEAC assay, Trolox equivalent antioxidant capacity test; TPA, 12-O-tetradecanoylphorbol-13-acetate; TR, thioredoxin reductase; TRAF6, TNF receptor associated factor 6; Trolox, 6-hydroxy-2,5,7,8-tetramethylchroman-2-carboxylic acid; Trx1, thioredoxin 1; UCP2, uncoupling protein 2; VEGF, vascular endothelial cell growth factor; XOD, xanthine oxidase; ZnPP, zinc protoporphyrin IX. Downward-pointing red arrows reflect the downregulatory action, upward-pointing green arrows reflect the upregulatory action.

### 3.1. Arylnaphthalene Skeletons

#### Sevanol

Sevanol, found in thyme of only one species *Thymus armeniacus* [117], was shown to possess acid-sensing ion channel (ASIC) inhibitory activity and a strong anti-inflammatory effect. Sevanol in vitro dose-dependently inhibited human and rat ASIC3 channels and, although with less efficiency, rat ASIC1a channels, heterologously expressed in oocytes of *Xenopus laevis*. In the model of neuronal-like cells, differentiated from the SH-SY5Y cell line by retinoic acid, sevanol showed an inhibitory effect on native human ASIC1a [118]. In Complete Freund’s Adjuvant (CFA)-induced thermal hyperalgesia test in vivo, sevanol showed an anti-inflammatory effect by significantly increasing the withdrawal latency on a hot plate and reducing edema of the inflamed hind paw [119,120,121]. It is intriguing that oral administration provides a more pronounced anti-inflammatory and analgesic effect, which indicates the appearance of a more active metabolite during metabolism [121].

### 3.2. Aryltetralin Skeletons

#### 3.2.1. Isoguaiacin

(–)-Isoguaiacin exerted diverse hepatoprotective activities by serving as a potent antioxidant. In primary cultures of rat hepatocytes injured withcarbon tetrachloride (CCl_4_), (–)-isoguaiacin significantly decreased the level of glutamic pyruvic transaminase (GPT), increased the level of reduced glutathione (GSH), decreased the production of malondialdehyde (MDA), a marker of lipid peroxidation, and preserved the activities of SOD, GPx and CAT [24].

#### 3.2.2. Isolariciresinol and Isolariciresinol Glucoconjugates

Isolariciresinol exhibited potent antioxidant activities in hydroxyl radical scavenging, 2,2′-diphenyl-1-picrylhydrazyl (DPPH) radical scavenging and Trolox equivalent antioxidant capacity (TEAC) in vitro assays, and inhibited ROS generation in HL-60 cells [25,26,27,28]. Isolariciresinol-9’-O-α-L-arabinofuranoside isolated from *Pinus massoniana* Lamb. exerted protective effects against oxidative stress-induced apoptosis in human umbilical vein endothelial cells (HUVECs) via a mechanism involving the upregulation of phosphatidylinositol 3-kinase (PI3K) and phosphorylated Akt, which lead to upregulated levels of phosphorylated Bcl-2-associated agonist of cell death (p-Bad) [31]. (+)-Isolariciresinol-3a-O-β-D-glucopyranoside extracted from *Carissa spinarum* exhibited moderate DPPH radical scavenging activity and high ferric reducing capacity (stronger than vitamin C), exerted significant hepatoprotective effects against H_2_O_2_-induced L02 cell injury by reducing ROS production, and also possessed a much better COX-2 inhibition activity compared with indomethacin [30]. (–)-Isolariciresinol-5-methoxy-9-β-D-xylopyranosyl showed the ability to scavenge DPPH radicals that were 1.5 times weaker than well-known antioxidant quercetin [29].

#### 3.2.3. Lyoniresinol and Its Derivatives

Lyoniresinol together with its glycated analog (+)-lyoniresinol-3α-O-β-glucopyranoside demonstrated significant capacity to scavenge DPPH radicals [33,34]. Lyoniresinol was also shown to effectively scavenge 2,2′-azino-bis(3-ethylbenzothiazoline)-6-sulfonic acid (ABTS) and hydroxyl radicals as well as reduce the intracellular ROS (iROS) in glutamate treated HT22 cells [32,33].

#### 3.2.4. Podophyllotoxin

Podophyllotoxin was found in plants of the genus Podophyllum, Linum, Callistris, and Juniperus. Due to its high toxicity and side effects, such as enteritis and depression of the central nervous system, the use of this compound is limited to a local antiviral agent. However, potent protective effects of podophyllotoxin formulation with rutin (G-003M) were demonstrated against radiation-induced lung injury. The formulation significantly attenuated oxidative and nitrosative stress and downregulated the expression of inflammatory and fibrogenic cytokines [122]. In another study, the efficacy of a more soluble and less toxic polyamidoamine dendrimer-conjugated podophyllotoxin was evaluated against chemically induced hepatocellular carcinoma (HCC) in mice. The administration of the drug significantly reduced histopathological changes in liver tissue and suppressed the progression of HCC by modulating the inflammatory and fibrogenic factors, which play important roles in HCC development [123].

#### 3.2.5. Sauchinone

Sauchinone, isolated from the root of *Saururus chinensis*, exerted anti-inflammatory function in vitro by suppressing NF-κB activity. Sauchinone was shown to dose-dependently inhibit the NF-κB-mediated production of NO and expression of inducible nitric oxide (NO) synthase (iNOS), TNFα, and COX-2 in LPS-stimulated RAW264.7 cells [124,125] and attenuated renal inflammation by inhibiting NF-κB/ROS pathway activation in angiotensin II (AngII)-induced human mesangial cells [35]. The lignan also showed anti-inflammatory functions in vivo in a murine model of allergen-induced airway inflammation. It suppressed neutrophil, lymphocyte, and eosinophil infiltration, and diminished pro-inflammatory cytokine production through the inhibition of GATA-3-driven T helper 2 (Th2) cell development, thereby attenuating tissue pathology [126].

**Table 2 ijms-23-06031-t002:** Antioxidant activity of lignans ex vivo and in vivo.

Lignan	Source	Model		Target	Dose, Road	Ref.
**Dibenzocyclooctadiene structure group**						
Gomisin A	*Schisandra chinensis* Baill.	CCl_4_-induced hepatotoxicity		↓MDA content, ↑SOD	50–100 mg/kg of rat, i.p.	[127]
Gomisin N	Synthetic	ethanol-injured model		↓iROS, ↑GSH/GSSG, ↑CAT, ↑SOD, ↑GPx, ↑SIRT1/AMPK, ↓CYP2E1	5–20 mg/kg of mice, p.o.	[38]
Schisandrin A	Synthetic	ovariectomy-induced osteoporosis		↓iROS level, ↑Nrf2	100 mg/kg of mice, i.p.	[43]
Schisandrin B	*Schisandra chinensis* (Turcz.) Baill.	I/R injury model		↑GSH/GSSG level	1.2 mmol/kg of rat, e.v.p.	[128]
	Synthetic	CCl_4_-induced hepatotoxicity		↑mtGSH/GSSG level, ↓mtMDA content, ↑GR, ↑GST, ↑GPx	2 mmol/kg of mice, p.o.	[129,130]
		ethanol-injured model		↓iROS, ↑GSH/GSSG level, ↑α-TOC, ↓MDA content, ↑GR, ↑GST, ↑MnSOD, ↑GPx	10 mg/kg of rat, i.g.	[131]
		Aβ-infused model		inhibition of ROO^•^-induced oxidation, ↑ORAC, ↑GSH/GSSG level, ↓MDA content, ↑SOD	25–50 mg/kg of rat, p.o.	[132]
		TSCI model		↑SOD	50 mg/kg of rat, p.o.	[133]
		I/R injury model		↓MDA content, ↑SOD	80 mg/kg of rat, p.o.	[134]
		STZ-induced diabetic model		↓iROS level, ↑Nrf2 ↑Bcl-2/Bax	20 mg/kg of mice, p.o.	[135]
		acute stress-induced anxiety		↓iROS level, ↓Keap1, ↑Nrf2, ↑SOD, ↑GSH/GSSG level	30–60 mg/kg of mice, p.o.	[136]
		pirarubicin-induced cardiotoxicity		↑SOD2, ↑CAT ↑Bcl-2/Bax	50 mg/kg of rat, diet	[137]
Schisandrin C	Synthetic	Ang II-induced endothelial deficit model		↓iROS level ↑Nrf2, ↑NQO-1, ↑HO-1, ↓Keap1	10 mg/kg of mice, i.g.	[138]
Schisantherin A	*Schisandra chinensis* (Turcz.) Baill.	Aβ-infused model		↓MDA content, ↑GSH/GSSG level, ↑SOD, ↑GPx	0.1 mg/kg of mice, i.c.v.	[139]
		chronic fatigue/D-galactose-induced LMI model		↑GSH/GSSG level, ↓MDA content, ↓Keap1, ↑Nrf2, ↑HO-1, ↑SOD, ↑CAT, ↑Bcl-2/Bax	2.5–5 mg/kg of mice, i.g.	[140,141]
	Synthetic	MCAO/R-induced brain injury		↓MDA level, ↑SOD, ↑Trx, ↑PRDx, ↓NOX4	5–10 mg/kg of rat, i.g.	[142]
**Dibenzylbutane structure group**						
Nordihydroguaiaretic acid	Synthetic	ozone-induced lung injury	↓tyrosine nitration level		20 mg/kg of rat, Alzet osmotic pumps	[61]
		K_2_Cr_2_O_7_-induced renal injury	↓NAG, ↑GPx		17 mg/kg of rat, mini-osmotic pumps	[143]
	*Larrea tridentata*	ALIOS-fed model	↑GPx4, ↑PRDx3, ↑PPARα		2.5 g/kg of mice, diet	[144]
Secoisolariciresinol diglucoside	*Linum usitatissimum* Linn.	metabolic syndrome model	↓TBARS, ↓iROS, ↑GSH/GSSG level, ↑SOD, ↑CAT, ↑GPx		20 mg/kg of rat, p.o.	[145]
	Synthetic	CCl_4_-induced hepato- and nephrotoxicity	↓MDA content, ↑CAT, ↑SOD, ↑POX, ↓LPO		12.5–25 mg/kg of rat, p.o.	[72]
		MCT-induced heart failure	↓iROS level, ↑SOD, ↑CAT, ↑GPx		25 mg/kg of rat, p.o.	[146]
	Synthetic (LGM2605)	CLP-induced sepsis	↓iROS level		100 mg/kg of mice, i.p.	[76]
		NRC painful model	↓8-OHG		200 mg/kg of rat, s.c.	[147]
	*Linum usitatissimum* Linn.	CdCl_2_-injured model	↑SOD, ↑CAT, ↑GPx, ↑GR		10 mg/kg of rat, s.c.	[148]
	Synthetic	BaP-injured model	↑GSH/GSSG, ↓MDA, ↑SOD, ↑CAT ↓p-p38, ↓p-ERK, ↑MKP-1, ↓miR-101A		100 mg/kg of mice, i.g.	[149]
		aging ovaries	↓iROS level		7–70 mg/kg of mice, i.g.	[150]
**Dibenzylbutyrolactone structure group**						
Arctigenin	*Arctium lappa* Linn.	WFST model		↑Nrf2, ↑SOD, ↑GR, ↑GPx, ↑Trx1, ↑UCP2, ↑p-AMPK, ↑p-p53, ↑p21, ↑PGC-1α, ↑PPARα	15 mg/kg of rat, i.p.	[80]
		ethanol-induced gastric ulcer		↓MDA content, ↑SOD	0.05–0.45 mg/kg of rat, p.o.	[151]
	Synthetic	JEV-infected model		↓ ROS level, ↑ SOD1	10 mg/kg of mice, i.p.	[152]
		LPS-injured model		↑GSH/GSSG level, ↓MDA content, ↑SOD, ↑CAT, ↑HO-1	50 mg/kg of mice, i.p.	[153]
		I/R injury model		↓MDA content, ↑SOD, ↑GPx ↑Nox1, ↑Trx1, ↑Nrf2	50–200 mg/kg of rat, i.g.	[154]
		AMI model		↓MDA content, ↑SOD, ↑GPx, ↑CAT, ↑HO-1	100–200 μmol/kg of rat	[155]
		Hep G2 xenograft model		↑p-p38, ↑p-JNK,↑Bax,↑TNF-α	20 mg/kg of mice, s.c.	[86]
		BLM-induced skin fibrosis		↑GSH/GSSG level, ↓MDA content, ↑SOD, ↑Nrf2, ↑HO-1	3 mg/kg of mice, i.p.	[156]
		I/R injury model		↓iROS level, ↓MDA content, ↑SOD ↑AMPK/SIRT1	100 μmol/kg of rat, i.p.	[87]
		cadmium-intoxicated model		↑GSH/GSSG, ↓8-oxo-dG level, ↓MDA, ↑GSR, ↑GCL, ↑GPx, ↑CAT ↑Nrf2, ↑HO-1, ↑NQO1	80 mg/kg of rat, i.g.	[157]
Hinokinin	Synthetic	HFD/STZ-induced type 2 diabetes		↓MDA, ↑SOD, ↑CAT, ↑GPx, ↑GST, ↑HO-1, ↑Nrf2, ↓Keap-1	20–40 mg/kg of mice, p.o.	[158]
Matairesinol	Synthetic	CLP-induced sepsis		↓MDA content, ↑SOD, ↑CAT, ↑GPx, ↑Nrf2, ↑HO-1, ↑AMPK	5–20 mg/kg of rat, p.o.	[93]
**Furofuranoid structure group**						
Fargesin	Synthetic	I/R injury model		↓MDA content, ↓ROS level, ↑SOD, ↑GPx, ↑CAT	15 μmol/kg of rat, i.v.	[159]
Isoeucommin A	*Eucommia ulmoides* Oliv.	H_2_O_2_-injured RTECs		↑SOD, ↑HO-1, ↑Nrf2	31.25–125 μM	[104]
				↑GSH/GSSG level	62.5–125 μM	
				↓MDA content	125 μM	
		STZ-induced diabetic nephropathy		↓MDA content, ↑GSH/GSSG level	2.5–10 mg/kg of rat, i.v.	
				↑SOD	5–10 mg/kg of rat, i.v.	
Pinoresinol diglucoside	Synthetic	Aβ-infused model		↓iROS level, ↓MDA content, ↑SOD, ↑CAT, ↑Nrf2, ↑HO-1 ↑Bcl-2/Bax	5–10 mg/kg of mice, i.g.	[160]
		MCAO model		↓iROS level, ↓MDA content, ↑GSH/GSSG level, ↑SOD, ↑GPx, ↑Nrf2, ↑NQO-1, ↑HO-1	5–10 mg/kg of mice, i.v.	[161]
Sesamin	Synthetic	STZ-induced diabetes		↓MDA content, ↑SOD	20 mg/kg of rat, p.o.	[162]
		nickel-induced hepatotoxicity		↓iROS, ↓TBARS, ↑GSH/GSSG level, ↓8-OHdG, ↑SOD, ↑CAT, ↑GPx ↑PI3K/AKT, ↑Bcl-2/Bax	60–120 mg/kg of mice, p.o.	[163]
		CCl_4_-induced hepatotoxicity		↓iROS, ↓TBARS level ↓p-JNK, ↓p-c-Jun, ↓cCYC, ↓Bax, ↓Bak, ↓Bcl-2	60–120 mg/kg of mice, p.o.	[164]
		fluoride-exposed model		↓iROS, ↓TBARS, ↑GSH/GSSG level, ↑SOD, ↑CAT, ↑GPx, ↑GST ↓p-JNK, ↓p-c-Jun, ↑Bcl-2/Bax	0.5–1 g/kg of carp, diet	[165]
	*Sesamum indicum* Linn.	DOX-treated model		↓iROS level, ↓MDA content ↑SOD, ↑CAT, ↑GPx	20–40 mg/kg of rat, i.g.	[166]
		6-OHDA model		↓iROS level, ↓MDA content, ↑SOD	20 mg/kg of rat, p.o.	[167]
		LPS-treated model		↑SOD, ↓MDA content	10 mg/kg of rat, p.o.	[168]
		LPS-treated model		↑GSH/GSSG level, ↓MDA content ↑SOD, ↑CAT, ↑Nrf2	100 mg/kg of mice, p.o.	[169]
		DSS-induced colitis		↓iROS level, ↑GSH/GSSG level, ↓MDA content, ↑SOD, ↑Nrf2, ↓Keap1, ↑HO-1, ↑NQO1, ↑GCLC, ↑GCLM, ↑GR, ↑p-AKT, ↑p-ERK1/2	50–100 mg/kg of mice, i.g.	[113]
		cisplatin-injured model		↓MDA content, ↑SOD, ↑Nrf2 ↓nitrate/nitrite ratio	5 mg/kg of rat, p.o.	[170]
		adult *Drosophila*		↑Nrf2/Cnc	2 mg/mL, diet	[171]
Syringaresinol	*Panax ginseng* C.A. Meyer	Sod1^–/–^ double-mutant model		↓iROS level, ↓8-isoprostane level ↓FoxO3a, ↓MMP-2	50 mg/kg of mice, p.o.	[172]
	*Sargentodoxa cuneata*	STZ-induced diabetes		↑Nrf2, ↑NQO-1, ↑HO-1, ↓Keap1, ↑SOD, ↑Bcl-2/Bax	25 mg/kg of mice, p.o.	[116]

6-OHDA, 6-hydroxydopamine; 8-OHG, 8-hydroxy-2-deoxyguanosine; α-TOC, α-tocopherol; Aβ, amyloid β; ALIOS model, American lifestyle-induced obesity syndrome model; AMI, acute myocardial infarction; Ang, angiotensin; BaP, benzo[a]pyrene; BLM, bleomycin; CLP, cecal ligation and puncture; Cnc, *Drosophila* Nrf2 orthologue; DOX, doxorubicin; DSS, dextran sulfate sodium; e.v.p., ex vivo pretreatment; HFD, high fat diet; i.c.v., intracerebroventricular administration; i.g., intragastric administration; i.p., intraperitoneal injection; i.v., intravenous injection; JEV, Japanese encephalitis virus; LMI, learning and memory impairment; LPO, lipid peroxidation; MCAO, middle cerebral artery occlusion model; MCT, monocrotaline; miR-101, microRNA 101a; MKP-1, protein-mitogen-activated protein kinase phosphatase 1; NAG, N-acetyl-β-D-glucosaminidase; NRC, nerve root compression; p.o., oral administration; RTECs, renal tubular epithelial cells; s.c., subcutaneous injection; STZ, streptozotocin; WFST, weight-loaded forced swimming test. Downward-pointing red arrows reflect the downregulatory action, upward-pointing green arrows reflect the upregulatory action.

### 3.3. Dibenzocyclooctadiene Skeletons

#### 3.3.1. Gomisins

Gomisin A ameliorated fibrogenesis and demonstrated hepatoprotective effect in the CCl_4_-induced acute liver injury model by suppressing the oxidative stress and activation of NF-κB. The treatment resulted in a decreased hepatic lipid peroxidation and increased SOD activity, as well as in the inhibition of pro-inflammatory mediators and iNOS [127]. Gomisin A protected against high glucose-induced oxidative stress in MC3T3 E1 cells via upregulation of potent antioxidant enzymes HO-1, copper-zinc SOD, manganese SOD and maintenance of mitochondrial homeostasis [36].

Gomisin N significantly increased the ROS leveland potentiated tumor necrosis factor-related apoptosis-inducing ligand (TRAIL)-induced apoptosis of HeLa cells through ROS-mediated up-regulation of death receptor 4 and 5, thereby demonstrating its potency in the treatment of malignant tumors [37].Gomisin N also exerted a protective effect against alcoholic liver disease by inhibiting hepatic steatosis, oxidative stress, and inflammation both in vitro in ethanol-treated male human Caucasian hepatocyte carcinoma (HepG2) cells and in vivo in ethanol-fed mice via the stimulation of hepatic sirtuin 1 (SIRT1)/AMP-activated protein kinase (AMPK) signaling. This was accompanied by downregulation of inflammation and lipogenesis, upregulation of fatty acid oxidation, and the suppression of cytochrome P450 2E1 (CYP2E1) followed by the enhancement of antioxidant genes and GSH levels in hepatic tissues [38].

#### 3.3.2. Schisandrins

Schisandrin A attenuated the increased ROS generation and the production of thiobarbituric acid reactive substance (TBARS), as well as prevented lipid peroxidation and enhanced the CYP3A4 mRNA level and protein activity in CCl_4_-treated HepG2 cells [39]. Different studies showed that schisandrin A inhibited NF-κB, c-Jun N-terminal kinase (JNK)/p38 MAPK, PI3K/Akt signaling pathways and activated the antioxidant Nrf2/HO-1 pathway. Schisandrin A decreased NO and prostaglandin E2 (PGE_2_) release, COX-2 and iNOS expression in a RAW 264.7 murine macrophage cell line, reduced plasma nitrite concentration in LPS-treated mice and attenuated xylene- induced ear edema and carrageenan-induced paw edema in vivo via the downregulation of the TLR4/NF-κB signaling pathway [173,174]. Additionally, schisandrin A showed a protective effect against LPS-induced inflammatory and oxidative responses in RAW 264.7 cells decreasing the expression of inflammatory mediators and cytokines, thereby diminishing the accumulation of iROS [40]. Schisandrin A protected the mitochondrial function in C2C12 skeletal muscle cells by eliminating the ROS under H_2_O_2_-induced oxidative stress [41]. Pre-treatment with schisandrin A protected human colorectal adenocarcinoma HT-29 cells against mycotoxin deoxynivalenol-induced cytotoxicity, oxidative stress and inflammation [42]. Schisandrin A suppressed the receptor activator of NF-κB ligand (RANKL)-induced osteoclastogenesis in vitro and prevented an ovariectomy-induced osteoporosis bone loss in vivo by reducing ROS production [43].

Schisandrin B is known as the main bioactive compound of *Schisandra chinensis* (Chinese magnoliavine), the plant of traditional Chinese medicine. Schisandrin B produces a variety of effects from apoptosis induction to anti-inflammatory and antioxidant action. Schisandrin B was shown to inhibit mitogen-induced phosphorylation of extracellular signal-regulated kinase (ERK), MAPK/ERK kinase (MEK), JNK, and p38, suppress IκBα degradation and nuclear translocation of NF-κB. All positive effects of schisandrin B were significantly reduced by Nrf2 and HO-1 inhibitors, which suggests that its anti-inflammatory effect was mediated by Nrf2 modulation [46].

Under oxidative stress, schisandrin B enhanced myocardial glutathione antioxidant status, thereby protecting against ischemia-reperfusion (I/R)-induced myocardial damage in isolated perfused rat hearts [128]. The cardioprotective effect of schisandrin B was also shownon H9c2 cardiomyocytes (rat embryonic cardiomyoblasts) in myocardial ischemia-reperfusion injury (MIRI) model through attenuation of the oxidative stress and inflammatory response via the AMPK/Nrf2 signaling pathway. Mechanistically, schisandrin B pretreatment reversed hypoxia/reoxygenation (H/R)-induced iROS generation, higher MDA content, upregulation of Keap1 and decreased enzymatic activities of SOD and GPx but induced the downregulation of pro-inflammatory cytokines (IL-1β, TNF-α and IL-8) and the upregulation of the anti-inflammatory cytokine IL-10 [50,51]. In another study, schisandrin B effectively protected the heart from injury caused by a doxorubicin analog pirarubicin by exerting strong antioxidant capacity [137].

In CCl_4_-induced hepatotoxicity in mice, schisandrin B catalyzed by hepatic P-450 triggered the enhancement of hepatic mitochondrial glutathione antioxidant status and induced heat shock responses in the liver [129,130,175]. In the model of long-term ethanol-treated rats, the treatment by schisandrin B reversed the altered mitochondrial antioxidant parameters, plasma reactive oxygen metabolites levels and mtMDA production in various tissues [131]. Oral administration of schisandrin B in the diabetic nephropathy mouse model significantly alleviated hyperglycemia-induced renal injury via the suppression of inflammatory response and oxidative stress [135].

Schisandrin B increased the resistance of dopaminergic cells to paraquat-induced oxidative stress and protected BJ human fibroblasts against solar irradiation-induced oxidative injury through the reduction in the oxidant-induced GSH depletion rate and the enhancement of the subsequent GSH recovery [44,45]. Schisandrin B prevented cyclosporine A-induced oxidative stress in human immortalized proximal tubular epithelial HK-2 cells and protected human keratinocyte-derived HacaT cells against t-butyl hydroperoxide-induced oxidative stress via scavenging ROS, increasing levels of mitochondrial membrane potential and GSH, promoting Nrf2 translocation into the nucleus followed by the target gene expression [48,49]. Schisandrin B treatment attenuated the vascular injury and fibrosis mediated by the endothelial to mesenchymal transition in vitro and in vivo [176]. Moreover, schisandrin B reduced epithelial cell injury in a model of colitis by modulating pyroptosis through AMPK/Nrf2/NLRP3 inflammasome pathways [177] as well as regulated STAT3-dependent Th17 cell differentiation and IL-17A cytokine release [178].

The neuroprotective potential of schisandrin B has been demonstrated in a number of experiments. It was found to protect nerve cells from apoptosis [47]. Another study presented schisandrin B as a neuroprotector for rats in a model of amyloid beta peptide (Aβ)-infused Alzheimer’s disease (AD), and revealed the potential role of schisandrin B for the cognitive improvement via the inhibition of the receptor for advanced glycation end products (RAGE)/NF-κB/MAPK axis [132]. Schisandrin B oral administration rescued the oxidative stress damage in amygdale and anxiety-like symptoms in forced swimming-induced anxiety model by upregulating Nrf2 expression and down-regulating Keap1 protein levels, reversing the SOD activity and GSH content and decreasing MDA and ROS levels in serum and amygdale [136]. In traumatic spinal cord injury (SCI) model of adult rats, schisandrin B also reversed the activation of injury-associated pathways, cancelling reduced SOD activity, increased MDA level and the activation of NF-κB p65 and TNF-α [133].

Schisandrin B potently suppressed lymphocyte activation, proliferation, and cytokine secretion in vitro, ex vivo and in vivo via alteration of cellular redox status. Schisandrin B enhanced the basal ROS levels, altered the GSH/GSSG ratio, induced nuclear translocation of Nrf2, and increased the transcription of corresponding genes in intact lymphocytes [46]. In another study, schisandrin B demonstrated anti-inflammatory and antioxidative effects in rat hind limb I/R skeletal muscle injury model [134]. Additionally, schisandrin B mitigated chondrocytes inflammation and ameliorated cartilage degeneration and osteoarthritis and attenuated hypoxia-induced inflammation, apoptosis and the progression of heart remodeling after myocardial infarction [179,180].

Importantly, schisandrin B can significantly reduce the viability of various cell lines in vitro at the concentrations of 40–100 μM [180,181] and cause hepatotoxicity in mice starting at a dose of 125 mg/kg [182], but was considered safe enough for animal treatment, since a dose of 80 mg/kg did not cause toxic effects [46].

Schisandrin C enhanced the resistance of cultured human fibroblasts to the solar irradiation-induced oxidative damage by eliciting the glutathione antioxidant response [45]. Orally administered schisandrin C ameliorated Ang II-induced oxidative stress in rat aortic endothelium cells (RAECs) by specifically binding to Keap1 and allowing for the Nrf2 translocation into the nucleus to promote the expression of its downstream antioxidant genes [138]. Schisandrin C also exhibited anti-inflammatory and antioxidant effects in human dental pulp cells (HDPCs). The effects were mediated by the upregulation of p-Akt-mediated Nrf2 pathway, the increase of the expression of PGC-1a and mitochondrial biogenesis. The MAPK pathway was downregulated, which was accompanied by the blocked NF-κB translocation to the nucleus and decreased production of ROS, NO, matrix metalloproteinase (MMP)-2/9, IL-1β, TNF-α, intracellular adhesion molecule-1 (ICAM-1) and vascular cell adhesion molecule-1 [52].

#### 3.3.3. Schisantherin A

Schisantherin A was shown to effectively inhibit lipid peroxidation and the production of TBARS, prevent the increased ROS generation and enhance the CYP3A4 mRNA level and protein activity in CCl_4_-treated HepG2 cells [39]. Generally, schisantherin A downregulated the expression of Keap1, Bax and caspase-3 and upregulated the expression of Nrf2, HO-1 and Bcl-2 which resulted in the increased level of GSH and decreased level of MDA [140,141] Intracerebroventricular (i.c.v.) administration of schisantherin A significantly attenuated Aβ-induced cognitive deficits, oxidative stress and neurodegeneration in the hippocampus and cerebral cortex which allows schisantherin A to be considered as a potential agent in the treatment of AD [139]. In another study, schisantherin A exerted neuroprotective effects and alleviated the symptoms of ischemic stroke, oxidative stress and inflammation responses in parietal cortex of rats after middle cerebral artery occlusion and reperfusion (MCAO/R) injury via the modulation of TLR4, NOX4, Trx1/Prx and C5aR1 signaling pathways [142]. Schisantherin A demonstrated antioxidative and anti-neuroinflammatory effects in LPS-activated BV-2 microglial cells [54] and could improve the learning and memory abilities of chronic fatigue mice and D-galactose treated mice [140,141]. Schisantherin A exerted a protective effect in human renal tubular epithelial cells subjected to H/R (model of renal I/R injury) [53] and also mitigated LPS-induced kidney inflammation via Nrf2-mediated NF-ĸB inhibition in rat tubular cells [55].

**Table 3 ijms-23-06031-t003:** Anti-inflammatory activity of lignans in vitro.

Lignan	Source	In Vitro Model	Target	Concentration	Ref.
**Arylnaphthalene structure group**					
Sevanol	*Thymus armeniacus*	HEO of *X. laevis*	↓hASIC3	IC_50_ 353 ± 23 μM	[119]
			↓rASIC1a	IC_50_ 2.2 ± 0.6 mM	
	Synthetic	HEO of *X. laevis*	↓rASIC3	IC_50_ 175 ± 18 μM	[120,121]
			↓rASIC1a	IC_50_ 227.5 ± 37.4 μM	
		RA-treated SH-SY5Y cells	↓hASIC1a	300 μM	[118]
**Aryltetralin structure group**					
(+)-Isolariciresinol 3a-O-β-D-glucopyranoside	*Carissa spinarum* Linn.	COX-2 assay	↓COX-2	IC_50_ 0.3 μM	[30]
Sauchinone	*Saururus chinensis*	LPS-stimulated RAW264.7	↓NO production	IC_50_ 4.08 μM	[124]
			↓iNOS, ↓TNF-α, ↓COX-2	1–30 μM	[125]
	Synthetic	AngII-induced mesangial cells	↓TGF-β,	0.1–1 μM	[35]
			↓NLRP3, ↓ICAM-1, ↓MCP-1, ↓IL-1β, ↓NF-κB p65	1 μM	
**Dibenzocyclooctadiene structure group**					
Schisandrin A	*Schisandra chinensis* (Turcz.) Baill.	LPS-stimulated RAW 264.7 macrophages	↓NO level, ↓iNOS, ↓PGE_2_, ↓COX-2, ↓NF-κB, ↑IκBα, ↓p-JNK, ↓p-p38 MAPK	25–100 μM	[173]
	Synthetic	LPS-stimulated RAW 264.7 macrophages	↓ iNOS, ↓ COX-2, ↓ TNF-α, ↓ IL-1β, ↑ IκB-α, ↓ p-JNK, ↓ p-p38 MAPK, ↓ p-ERK, ↓ p-PI3K, ↓ p-Akt	200 μM	[40]
		DON-induced cytotoxicity in HT-29 cells	↓ PGE_2_, ↓ COX-2, ↓ NF-κB, ↓ IL8, ↓ p-p38, ↓ p-ERK	2.5–10 μM	[42]
		RANKL-induced osteoclast differentiation	↓ PGE_2_, ↓ COX-2, ↓ NF-κB, ↓ IL8, ↓ p-p38, ↓ p-ERK	50–200 μM	[43]
Schisandrin B	Synthetic	Con A-induced lymphocytes	↓ NF-κB, ↓ p-MEK, ↓ p-p38, ↓ p-ERK, ↓ p-JNK, ↑ IκBα, ↓ IL-2, ↓ IL-4, ↓ IL-6, ↓ IFN-γ	25–50 μM	[46]
		Ang II/TNF-α/ROS-induced HUVECs	↓NF-κB, TNF-α, ↓p-Smad2/3, ↓vimentin, ↓α-SMA, ↓Snail/slug, ↓TGF-β, ↓Twist, ↑VE-cadherin	10 μM	[176]
		TH17 cell differentiation	↓p-STAT3	1 μM	[178]
		H/R-induced H9c2 cell injury	↓ IL-1β, ↓ TNF-α, ↓ IL-6, ↓ IL-8, ↓ TGF-β, ↑ IL-10	20 μM	[50,51]
		LPS+ATP-treated intestinal epithelial cells	↓TNF-α, ↓IL-6, ↓IL-18, ↓IL-1β, ↓NLRP3, ↑p-AMPK	40 μM	[177]
Schisandrin C	Synthetic	LPS-stimulated HDPCs	↓ NO level, ↓ p-ERK1/2, ↓ p-SAPK/JNK, ↓ p-p38, ↓ NF-κB	10–20 μM	[52]
Schisantherin A	*Schisandra chinensis* (Turcz.) Baill.	H/R-induced HK-2 cells	↓ TNF-α, ↓ IL-1β, ↓ IL-6	5–20 μM	[53]
		LPS-stimulated BV-2 microglial cells	↓ NF-κB, ↓ IKK, ↑ IκB, ↓ TNF-α, ↓ IL-6, ↓ IL-1β, ↑ IL-10 ↓ iNOS, ↓ COX-2 ↓ p-p38, ↑ p-ERK, ↓ p-JNK, ↓ p-Akt	50 μM	[54]
	Synthetic	LPS-stimulated NRK-52E cells	↓NF-κB, ↓TNF-α, ↓Rantes	25–50 μM	[55]
**Dibenzylbutane structure group**					
Nordihydroguaiaretic acid	Synthetic	IL-1β-induced PC12 cells	↓APP secretion and processing	10 μM	[183]
		IFN-γ- induced rat brain astrocytes/C6 cells	↓IRF-1, ↓IP-10 ↓p-STAT1, ↓p-STAT3,↓p-↓JAK2	5–20 μM	[184]
		RANKL-induced bone marrow-derived macrophage/RAW-D cells	↓ osteoclast differentiation, ↓ RANKL-induced signal cascade ↓ NFATc1, ↓ p-ERK	1–10 μM	[185]
Secoisolariciresinol diglucoside	*Linum usitatissimum* Linn.	iron treated H9c2 cells	↓ TNF-α, ↑ IL-10	500 μM	[73]
		CdCl_2_-injured model	↓MPO, ↓NO level	10 mg/kg of rat	[148]
	Synthetic (LGM2605)	asbestos-exposed MFs	↓iNOS, ↓IL-1β, ↓IL-6, ↓IL-18, ↓TNFα	50–100 μM	[74,75]
**Dibenzylbutyrolactone structure group**					
Arctigenin	*Forsythia fructus*	pro-inflammatory enzyme assays	↓ PLA2, ↓ COX-1, ↓ COX-2, ↓ 5-LOX	100 μM	[186]
	*Arctium lappa* Linn.	bone marrow-derived MDSCs	↑Arg-1, ↑iNOS	10–20 μM	[187]
	Synthetic	LPS-treated Raw264.7 cells	↓ iNOS, ↓ p-STAT, ↓ IL-1β, ↓ IL-6, ↓ MCP-1, ↓ p-JAK2	5–50 μM	[78]
		TGF-β1-induced HK-2 cells	↓ NF-κB p65, ↓ MCP-1	0.5–1 μM	[83]
		OA-treated WRL68 hepatocytes	↓ ICAM-1, ↓ IL-1β, ↓ IL-6, ↓ IL-7, ↓ IL-8, ↓ TNFα	50 μM	[85]
		LPS-treated RAW264.7 cells	↓TNF-α, ↓IFN-γ, ↓IL-17, ↓IL-1β, ↓CXCL10, ↑TGF-β1, ↑IL-4	10–100 μM	[188]
		LPS-treated RAW264.7 cells	↓TNF-α, ↓IFN-γ, ↓IL-17, ↓IL-1β, ↓CXCL10, ↑TGF-β1, ↑IL-4	10–100 μM	[188]
		IL-1β–stimulated human chondrocytes	↓ TNF-α, ↓ COX-2, ↓ iNOS, ↓ IL-6, ↓ PGE_2_, ↓ NO, ↑ IκBα, ↓ p65, ↓ PI3K, ↓ Akt	10–50 μM	[189]
		scintillation proximity assay	↓PDE4	IC_50_ 3.76 ± 0.28 μM	[190]
		LPS-stimulated human PBMCs	↓ TNF-α	IC_50_ 35.18 ± 6.01 μM	
		LPS-treated RAW264.7 cells	↓TNF-α, ↑p-CREB, ↓PDE4	100 μM	
		OGD-injured H9c2 cardiomyocytes	↓NF-κB, ↑IKBα, ↓TNF-α, ↓IL-1β, ↓IL-6	50–200 μM	[87]
		silica-injured RAW 264.7 macrophages	↓iNOS, ↓Arg-1, ↓TLR-4, ↓NLRP3, ↓TGF-β	1 μM	[88]
Hinokinin	*Aristolochia indica* L.	LPS-stimulated THP-1 cells	↓ IL-6	20.5 ± 0.5 μM	[191]
			↓ TNF-α	77.5 ± 27.5 μM	
Matairesinol	Synthetic	naive CD4^+^ T cells	↓p-p38, ↓p-ERK, ↓ROR-γt	20 μM	[192]
		LPS-stimulatedNSC-34neurons andBV2microglia	↓TNF-α, ↓IL-1β, ↓IL-6, ↓IFN-γ, ↓IL-8, ↓MCP1, ↓MAPK, ↓JNK, ↓NF-κB	5–20 μM	[93]
Matairesinol-7′-hydroxyl	*Piceaabies*	TNF-α-induced HAEC	↓ICAM-1, ↓VCAM-1, ↓monocyte adhesion	0.1–100 μM	[193]
			↓p-NF-κB	10–100 μM	
			↓p-ERK	100 μM	
Nortrachelogenin	Synthetic	LPS-stimulated J774 macrophages	↓PGE_2_, ↓NO, ↓iNOS	1–30 μM	[194]
			↓MCP-1, ↓IL-6	3–30 μM	
			↓mPGES-1	30 μM	
**Furanoid structure group**					
(−)-Olivil	*Osmanthus fragrans* var. *aurantiacus*	LPS-activated RAW264.7 cells	↓NO level	IC_50_ 85.6 ± 1.49 μM	[195]
Taxiresinol	*Osmanthus fragrans* var. *aurantiacus*	LPS-activated RAW264.7 cells	↓NO level	IC_50_ 58.1 ± 1.42 μM	[195]
	*Perovskiaatriplicifolia* Benth	RBL-1 leukemia cells	↓leukotriene C4 release	IC_50_ 3.4 ± 0.09 μM	[196]
**Furofuranoid structure group**					
Dendranlignan A	*Dendranthema morifolium* (Ramat.)	LPS-induced H9c2 cells	↓TNF-α, ↓IL-6, ↓IFN-γ ↓p-cJUN, ↓p-P65, ↓p-IRF3	10 μM	[103]
(+)-Diayangambin	*Piper fimbriulatum*	human mononuclear cells	↓proliferation	1.5 μM	[197]
		LPS-stimulated RAW264.7 macrophages	↓ PGE_2_	10 μM	
Fargesin	*Magnolia fargesii*	PMA-stimulated THP-1	↓iNOS, ↓COX-2, ↓IL-1β, ↓TNF-α, ↓AP-1, ↓NF-κB, ↓JNK	5–20 μM	[198]
	*Magnolia* sp.	LPS-stimulated RAW264.7	↓iNOS, ↓COX-2, ↓NF-κB	25 μM	[199]
Koreanaside A	*Forsythia koreana*	LPS-stimulated RAW 264.7 macrophages	↓iNOS, ↓COX-2, ↓IL-6, ↓TNF-α, ↓p-IκBα, ↓p-TAK1	20–80 μM	[200]
			↓AP-1, ↓p-c-Fos, ↓p-p65, ↓NF-κB, ↓p-IKKα/β, ↓p-STAT1, ↓p-STAT3, ↓p-JAK1, ↓p-JAK2	40–80 μM	
Phillygenin	*Forsythia koreana*	RAW 264.7 cells	↓PGE_2_, ↓NO, ↓iNOS, ↓NF-κB	1–100 μM	[201]
Pinoresinol	Synthetic	IL-1β-stimulated Caco-2 cells	↓ PGE_2_, ↓ MCP-1, ↓ NF-κB	50–100 μM	[202]
			↓IL-6	10–100 μM	
Pinoresinol diglucoside	*Eucommia ulmoides*	oxLDL-induced HUVEC cytotoxicity	↓ eNOS, ↓ p-p38MAPK, ↓ p-NF-κB p65	1 μM	[109]
Sesamin	*Sesamum indicum* Linn.	oxLDL-induced HUVECs cytotoxicity	↓NF-κB, ↓IL-8	12.5–100 μM	[110]
		FPR-transfected ETFR cells, THP1 cells	↓cell migration, ↓NF-κB activation, ↓ERK1/2 phosphorylation	6.25–50 μM	[203]
		KA-induced PC12 and BV-2 cells	↓ERK1/2, ↓p38 MAPK, ↓COX-2	10–50 μM	[111]
		RPMC	↓histamine release	25–100 μM	[204]
		HMC-1	↓TNF-α, ↓IL-6, ↓p38 MAPK, ↓NF-κB		
	Synthetic	RLE-6TN and L2 cells	↑A20, ↑TAX1BP1	10 μM	[205]
*epi*-Sesamin	*Asarum siebodlii*	HUVEC	↓EPCR shedding	1–10 μM	[206]
Syringaresinol	*Perovskiaatriplicifolia* Benth	RBL-1 leukemia cells	↓leukotriene C4 release	IC_50_ 7.9 ± 0.04 μM	[196]
	*Rubia philippinensis*	LPS-stimulated RAW 264.7 cells	↓iNOS, ↓COX-2, ↓TNF-α, ↓IL-1β, ↓IL-6, ↓PGE_2_, ↓ERK1/2, ↓JNK, ↓p38 MAPK	25, 50, 100 μM	[207]
		High glucose-treated NRVM	↓TNF-α, ↓IL-6, ↓IL-1β, ↓TGF-β, ↓p-Smad2/3	50, 100 μM	[116]
		LPS^+^ ATP-treated H9c2 cells	↓IL-1β, ↓IL-18, ↑SIRT1 expression, ↓NLRP3 inflammasome activation	100 μM	[208]
	*Albiziae cortex*	BV2 microglia cells	↓TNF-α, ↓IL-6, ↓IL-1β, ↓COX-2, ↓NO, ↑M2 phenotype, ↓NF-κB	25, 50, 100 μM	[209]

AKT, protein kinase B; APP, amyloid precursor protein; ASIC 1/3 h/r, acid-sensing ion channel isoform 1/3 type human/rat; BV-2 cells, murine microglial cell line; COX-2, cyclooxygenase-2; CREB, cAMP-response element binding protein; DON, trichothecene toxin deoxynivalenol; EPCR, endothelial protein C receptor; ERK, extracellular signal-regulated kinase; ETFR, epitope-tagged human FPR cell; FPR, formyl peptide receptor; H9c2 cells, embryonic rat heart derived cell line; HAEC, human aortic endothelial cells; HDPCs, human dental pulp cells; HEO of *X. laevis*, heterologously expressing oocytes of *Xenopus laevis* frog; HK-2, human renal tubular epithelial cells; HMC-1, human mast cell line 1; H/R, hypoxia/reoxygenation; HUVECs, human umbilical vein endothelial cells; IκBα, inhibitor of κB; IKKα/β, IκB kinase; IL, interleukin; iNOS, inducible nitric oxide synthase; IP-10, inducible protein-10; IRF-1, interferon regulatory factor-1; JAK2, Janus kinase 2/signal transducer; JNK, c-Jun N-terminal kinase; KA, kainic acid; LPS, lipopolysaccharide; MAPK, mitogen-activated protein kinase; mPGES-1, microsomal prostaglandin E synthase-1; MPO, myeloperoxidase; NF-κB, nuclear factor-κB; NLRP3, nod-like receptor family pyrin domain containing 3; NRK-52E cells, normal rat kidney cell line; NRVM, neonatal rat ventricular myocytes; OA, oleic acid; OGD, oxygen glucose deprivation; oxLDL, oxidized low-density lipoprotein; PBMCs, human peripheral blood mononuclear cells; PC12 cells, rat pheochromacytoma cell line; PDE, phosphodiesterase; PGE2, prostaglandin E2; PMA, phorbol-12-myristate-13-acetate; PPAR, peroxisome proliferator-activated receptor; RA, retinoic acid; RANKL, receptor activator of NF-κB ligand; RLE-6TN, rat lung epithelial-6-T-antigen negative cell line; ROR-γt, retinoid-related orphan receptor-γt; RPMC, rat peritoneal mast cell; SH-SY5Y, human neuroblastoma cell line; SIRT1, NAD-dependent deacetylase sirtuin-1; STAT3, signal transducer and activator of transcription 3; TAK1, TGF-β-activated kinase 1; TGF-𝛽, transforming growth factor-𝛽; TNF-α, tumor necrosis factor α. Downward-pointing red arrows reflect the downregulatory action, upward-pointing green arrows reflect the upregulatory action.

### 3.4. Dibenzylbutane Skeletons

#### 3.4.1. Nordihydroguaiaretic Acid (NDGA) and Its Derivatives

NDGA, found in leaves and twigs of the evergreen desert shrub *Larrea tridentata* (Sesse and Moc. ex DC) Coville (creosote bush), is the best known lignan with lipoxygenase-inhibitory activity [210]. However, many studies also demonstrate its antioxidant and anti-inflammatory properties. NDGA showed potent antioxidant activity against iROS in HL-60 cells [59]. It was shown that NDGA scavenges efficiently at least two hydroxyl radicals (HO^•^) generated by the Fenton reaction [63]. NDGA also demonstrated scavenging effects against ONOO^–^ peroxynitrite anion as efficiently as uric acid; against ^1^O_2_ singlet oxygen more efficiently than dimethyl thiourea, lipoic acid, N-acetyl-cysteine (NAC) and glutathione; against OH^•^ hydroxyl radicals more efficiently than dimethyl thiourea, uric acid, Trolox, dimethyl sulfoxide and mannitol; against O_2_^•–^ superoxide anion more efficiently than NAC, glutathione, tempol and deferoxamine; against HOCl hypochlorous acid as efficiently as lipoic acid and NAC; and was unable to scavenge H_2_O_2_. It should be noted that NDGA exerted protective effects not only by scavenging ROS but also by inducing the expression of cytoprotective genes. NDGA prevented iROS accumulation and mitochondrial depolarization in Keap1-independent manner via the inhibitory phosphorylation of glycogen synthase kinase-3 (GSK-3) and the activation of ERK1/2, p38, JNK, and PI3K pathways with a subsequent nuclear localization of Nrf2 and activation of the ARE regulatory sequence and the increase in HO-1 protein levels [65].

In an in vivo study in rats, NDGA was able to prevent ozone-induced tyrosine nitration in lungs [61]. NDGA treatment ameliorated oxidative and nitrosative stress and showed renoprotective effect in a K_2_Cr_2_O_7_-induced nephrotoxicity model in rats [143]. NDGA protected cerebellar granule neurons against H_2_O_2_- or 3-nitropropionic acid-induced neurotoxicity [62]. In kidney-derived LLC-PK1 and HEK293T cells and in wild-type mouse embryo fibroblasts (MEFs), NDGA prevented H_2_O_2_-induced cell death [65]. NDGA attenuated toxicity, ROS production and the oxidative stress-induced decrease of CD33 (a myeloid cell-specific type I transmembrane glycoprotein) expression in the iodoacetate- or H_2_O_2_-treated human acute monocytic leukemia (THP-1) cell line [66].

Dietary administration of NDGA to obese mice improved the metabolic disregulation by upregulating PPARα, hepatic antioxidant enzymes, GPx4, mitochondrion-specific antioxidant peroxiredoxin 3, and expression of key genes involved in fatty acid oxidation together with downregulating the key lipogenic enzymes, apoptosis and ER stress signaling pathways [144].

In the model of mouse skin treated by stage I tumor promoting agent, 12-O-tetradecanoylphorbol-13-acetate, NDGA pretreatment mitigated cutaneous lipid peroxidation, inhibited H_2_O_2_ production, restored reduced GSH level and activatied antioxidant enzymes, lowered the elevated activities of myeloperoxidase (MPO), xanthine oxidase and skin edema formation, thereby demonstrating antioxidative and anti-inflammatory properties and chemopreventive potential against skin cancer [64].

Furthermore, NDGA inhibited the inflammatory response after SCI by decreasing MPO, IL-1β and TNF-α levels and the number of macrophages/microglia, thereby limiting secondary damage and demonstrating neuroprotective potential [211]. In cultured rat brain astrocytes, NDGA suppressed IFN-γ-induced inflammatory responses by inhibiting JAK/STAT activation and downregulating the inflammatory mediators IRF-1 and IP-10 [184]. NDGA inhibited the IL-1β-increased maturation, processing and secretion of amyloid precursor protein in PC12 cells, thereby indirectly contributing to the attenuation of the amyloid plaque formation in AD [183].

NDGA showed ameliorating potential on inflammatory bone destruction mediated by osteoclasts via the inhibition of calcium oscillation followed by the downregulation of a key transcription factor for osteoclastogenesis NFATc1 and inhibition of RANKL-induced osteoclastogenesis in cultures of murine osteoclast precursor cell line RAW-D and primary bone marrow-derived macrophages [185]. Dietary supplementation with NDGA ameliorated dyslipidemia and hepatic steatosis in *ob/ob* mice via PPARα-dependent and PPARα-independent lipid pathways and AMPK signaling [212]. In cecal ligation and double puncture (CLP)-induced abdominal sepsis model in rats, NDGA treatment improved oxygenation, decreased lactate, lowered lung injury and mitigated lung edema, thereby demonstrating its anti-inflammatory potential in the modulation of organ injury [213].

Aside its protective effects, NDGA has also shown cytotoxic effects in several studies. NDGA treatment led to the increase in oxidative processes, phosphorylation and activation of the MAP kinases ERK, JNK and p38, causing apoptosis of murine pro-B lymphocytes (FL5.12 cells) [60], and evoked cell death inducing oxidative damage of proteins in the medulloblastoma-derived Daoy cell line [67]. Thus, NDGA is not a safe compound. The Food and Drug Administration (FDA) removed NDGA from the FDA’s list of Generally Regarded as Safe (GRAS) agents as early as 1968. NDGA was shown to cause cystic nephropathy in rats and skin hypersensitivity in humans, and high doses of *L. tridentata* were associated with kidney disease and hepatotoxicity in humans [214,215]. The LD_50_ of NDGA was found to be 75 mg/kg (i.p. administration in mice) with higher NDGA doses (100 and 500 mg/kg) causing 100% mortality within 30 h [214].

#### 3.4.2. Meso-Dihydroguaiaretic Acid

Meso-dihydroguaiaretic acid showed hepatoprotective activity for rat hepatocytes that manifested in a significant decrease of GPT level released into the medium from the primary culture. It also preserved the GSH/GSSG and MDA levels, and SOD, GPx, CAT activities [24]. Meso-dihydroguaiaretic acid demonstrated strong free radical scavenging activity in various cell-free assays, modulated MAPKs/Akt signal pathways in G-protein coupled receptor agonists-induced human neutrophils, and reduced ROS generation. In a murine model of LPS-induced acute respiratory distress syndrome, the lignan application showed anti-neutrophilic inflammatory effects [58].

#### 3.4.3. Secoisolariciresinol and Its Derivatives

Secoisolariciresinol exhibited strong scavenging activity against the stable free radical DPPH [56,68,70,71] and potently scavenged superoxide and peroxyl radicals, being more effective than two generally accepted standards: butylated hydroxyanisole (BHA) and Trolox [69]. Secoisolariciresinol also showed strong protection against AAPH peroxyl radical that is capable to initiate plasmid DNA nicking and phosphatidylcholine liposome lipid peroxidation [71]. 7-Hydroxy-Secoisolariciresinol also showed significant but weaker DPPH radical scavenging activity than secoisolariciresinol [70].

Secoisolariciresinol diglucoside (SDG) exerted high in vitro antioxidant potency to DPPH and AAPH scavenging similar to unglicosilated secoisolariciresinol [71,72]. In various models of pathological states, SDG showed cytoprotective effect by reversing AMPK and mitogen-activated protein kinase phosphatase 1 (MKP-1) expression that restored the activity of antioxidant enzymes (SOD, CAT, GPx, GR, and peroxidase (POX)). It decreased pro-apoptotic protein levels, suppressed pro-inflammatory signaling (p-p38 MAPK, p-ERK, NF-κB) and the expression of inflammatory mediators (TNF-α, IL-10, interferon γ (IF-γ), MMP-2/9) [149,150]. SDG showed cytoprotective effect in cardiac iron overload-induced redox-inflammatory damage condition suggesting the cardioprotective role for this flaxseed lignan [73].

SDG improved ovarian aging by inhibiting oxidative stress and scavenging slowly accumulated ROS in ovarian tissues [150]. In in vivo models of CCl_4_ and benzo[a]pyrene intoxication, SDG attenuated oxidative damage in liver and kidney tissues [72]. Additionally, SDG protected kidneys from cadmium-induced oxidative damage by restoring antioxidant enzymes activity and decreasing lipid peroxidation [148,149]. SDG showed protective effect against oxidative stress in rats with metabolic syndrome by preventing lipids from oxidative damage, improving enzymatic antioxidant defenses and GSH level [145]. Pre-treatment with SDG provided protection in a monocrotaline-induced model of pulmonary arterial hypertension by decreasing right ventricle hypertrophy, ROS levels, lipid peroxidation, plasma levels of alanine transaminase and aspartate transaminase [146].

A number of studies have been carried out on LGM2605 that is a chemically synthesized SDG by a proprietary route. It was shown that LGM2605 treatment reduced 8-hydroxy-2-deoxyguanosine (8-OHdG), attributed to ROS-specific nuclear damage, and nitrotyrosine in DRG and spinal neurons of rats after a painful nerve root compression [147]. As it was shown in human ventricular cardiomyocyte-derived cell line AC16 treated with LPS and CLP mouse model of peritonitis-induced sepsis, LGM2605 alleviated oxidative stress, increased mitochondrial respiration, and restored cardiac systolic function by directly decreasing ROS accumulation not via affecting the expression of antioxidant genes but preventing the activation of NF-κΒ [76]. LGM2605 treatment significantly mitigated asbestos-induced cytotoxicity, inflammation, and oxidative damage by reducing ROS generation and nitrosative stress, decreasing levels of MDA and 8-iso Prostaglandin F2α (8-isoP) (markers of lipid peroxidation), enhancing Nrf2 activation and the expression of phase II antioxidant enzymes, HO-1 and Nqo1, as well as reducing levels of IL-1β, IL-18, IL-6, and TNFα in both WT and Nrf2^−/−^ murine peritoneal macrophages, supporting its possible use as a chemoprevention agent in the development of asbestos-induced malignant mesothelioma [74,75].

**Table 4 ijms-23-06031-t004:** Anti-inflammatory activity of lignans ex vivo and in vivo.

Lignan	Source	Model	Target	Dose, Road	Ref.
**Arylnaphthalene structure group**					
Sevanol	*Thymus armeniacus*	CFA-induced thermal hyperalgesia	↑withdrawal latency of inflamed hind paw	1–10 mg/kg of mice, i.v.	[119]
	Synthetic	CFA-induced paw edema	↓paw edema	0.1–1 mg/kg of mice, i.m., i.n., p.o.	[120,121]
**Aryltetralin structure group**					
Podophyllotoxin	Synthetic (G-003M)	TGR-exposed model	↑survival, ↓NO, ↓IL-6, ↓TNF-α, ↓TGF-β1	5 mg/kg of mice, i.m.	[122]
	Synthetic, conjugated with PAA dendrimer	HCC-induced model	↓IL-6, ↓NF-κB, ↓α-SMA, ↓TGF-β	10, 20 mg/kg of mice, p.o.	[123]
Sauchinone	*Saururus chinensis*	OVA-induced asthma model	↓neutrophil, lymphocyte, eosinophil infiltration in BALF, ↓IL-5, ↓IL-13, ↓Th2 cell development	10, 100 mg/kg of mice, i.p.	[126]
**Dibenzocyclooctadiene structure group**					
Gomisin A	*Schisandra chinensis* Baill.	CCl_4_-induced hepatotoxicity	↓TNF-α, ↓IL-1β, ↓iNOS, ↓NF-κB, ↓p-IκB	50–100 mg/kg of rat, i.p.	[127]
Gomisin N	*Schisandra chinensis* Baill.	ethanol-induced liver injury	↓ NF-κB p65, ↑ IκB, ↓ TNF-α, ↓ IL-6, ↓ MCP-1	5–20 mg/kg of mice, p.o.	[38]
Schisandrin A	*Schisandra chinensis* (Turcz.) Baill.	LPS-treated model	↓NO level	100–200 mg/kg of mice, i.p.	[173]
		carrageenan-induced paw edema	↓paw edema volume		
		xylene-induced ear edema	↓ear edema degree	25–50 mg/kg of mice, p.o.	[174]
		carrageenan-induced paw edema	↓paw edema volume ↓TNF-α, ↓IL-1β, ↓MPO, ↓p-p65NF-κB, ↓p-IκB, ↓TLR4	25–50 mg/kg of mice, p.o.	
Schisandrin B	Synthetic	Con A-induced lymphocytes	↓IL-2, ↓IL-4, ↓IL-6, ↓IFN-γ	80 mg/kg of mice, i.p.	[46]
		myocardial infarction model	↓in left ventricular end-systolic and end-diastolic diameter, ↓heart weight/body weight ratio, ↓infarct size, ↓NF-κB, ↓TGF-β1, ↓TNF-α	80 mg/kg of mice, i.g.	[179]
		Aβ-infused model	↓ COX-2, ↓ iNOS, ↓ TNF-α, ↓ IL-1β, ↓ IL-6	25–50 mg/kg of rat, i.g.	[132]
		I/R injury model	↓ IL-1β, ↓ TNF-α, ↓ p-p38MAPK, ↓ p-ERK1/2, ↓ NF-κB p65	80 mg/kg of rat, p.o.	[134]
		TSCI model	↓ NF-κB p65, ↓ TNF-α	50 mg/kg of rat, p.o.	[133]
		IL-1β-induced rat chondrocytes	↓IL-6, ↓iNOS, ↓MMP3, ↓MMP13, ↓NF-κB, ↓MAPK	50 μM, i.a.	[180]
		STZ-induced diabetes	↑ IκBα, ↓ VCAM-1, ↓ TNF-α	20 mg/kg of mice, p.o.	[135]
		Ang II-induced vascular injury model	↓α-SMA, ↓p-Smad2/3, ↑VE-cadherin	20 mg/kg of mice, p.o.	[176]
		DSS induced colitis	↓TNF-α, ↓IL-6, ↓IL-18, ↓IL-1β, ↓NLRP3, ↑p-AMPK	10 mg/kg of mice, i.p.	[177]
Schisantherin A	Synthetic	MCAO/R-induced brain injury model	↓IL-1β, ↓IL-6, ↓p-IκBα, ↓NF-κB, ↓p-ERK, ↓p-JNK, ↓p-p38, ↓TLR4, ↓C5aR1	5–10 mg/kg of rat, i.g.	[142]
		LPS-induced acute kidney injury	↓accumulation neutrophils and T-lymphocytes, ↓NF-κB, ↓TNF-α, ↓Rantes	40 mg/kg of mice, i.p.	[55]
**Dibenzylbutane structure group**					
Meso-dihydroguaiaretic acid	*Machilusphilippinensis* Merr.	LPS-induced ARDS model	↓MPO, ↓4-HNE, ↓elastase accumulation	30 mg/kg of mice, i.p.	[58]
Nordihydroguaiaretic acid	Synthetic	TPA-treated model	↓LPO level, ↓XOD, ↓MPO	15–25 μM, shaved area of dorsal skin	[64]
		spinal cord injury	↓MPO, ↓TNF-α, ↓IL-1β	30 mg/kg of rat, i.p.	[211]
		leptin-deficient (*ob/ob*) mice	↑PPARα, ↑p-AMPK ↑fatty acid oxidation pathway	0.83 g/kg, 2.5 g/kg, diet	[212]
		CLP-induced sepsis	↓lung edema, ↓lactate, ↓blood urea nitrogen, ↓histologic lung injury	20 mg/kg of rat, i.p.	[213]
Secoisolariciresinol diglucoside	Synthetic	CLP-induced sepsis	↓p-IκBα, ↓NF-κΒ	100 mg/kg of mice, i.p.	[76]
		BaP-injured model	↓MPO, ↓NO level, ↓TNF-α, ↓IL-6, ↓IL-1β, ↓NF-κB	100 mg/kg of mice, i.g.	[149]
**Dibenzylbutyrolactone structure group**					
Arctigenin	*Forsythia fructus*	OVA-induced asthma model	↓ PDE	10–100 μM	[186]
		48/80-induced RPMCs	↓ histamine release	10 μM	
		IgE-rich mouse serum-induced PCA skin model	↓ amount of Evans blue leakage	15–45 mg/kg of rat, p.o.	
		anti-rat rabbit serum antibody-induced RCA skin model	↓ skin edema		
		SRBC-induced Arthus reaction model	↓footpad thickness, ↓hemolysis tier, ↓hemagglutinin titer, ↓plaque-forming cells	15–45 mg/kg of mice, p.o.	
		SRBC-induced DTH model	↓ footpad thickness, ↓ rosette-forming cells		
		DNFB/PC-induced contact dermatitis	↓ ear edema	0.1–1 mg/ear of mice	
	*Arctium lappa* Linn.	LPS-/PGN-stimulated peritoneal macrophages	↓IL-6, ↓TNF-α, ↓IL-1β, ↑IL-10, ↑CD204, ↓p-PI3K, ↓p-Akt, ↓p-p65, ↓p-IKKβ	10–20 μM	[216]
		LPS-/PGN-induced model	↓TNF-α, ↓IL-1β	5 mg/kg of mice, i.p.	
		TNBS-induced colitic model	↓IL-6, ↓TNF-α, ↓IL-1β, ↓MPO, ↑IL-10, ↓p-PI3K, ↓p-Akt, ↓p-p65	30–60 mg/kg of mice, p.o.	
		acetic acid-induced chronic ulcer model	↓ TNF-α, ↓ IL-6, ↑ IL-10, ↓ CRP	0.05–0.45 mg/kg of rat, p.o.	[151]
		LPS-induced acute inflammation model	↓CD86, ↓IL-6, ↓IL-12, ↓TNF-α, ↓IL-1β, ↑IL-10, ↑G-MDSCs, ↓M-MDSCs, ↓IRF8, ↑miR-127-5p, ↓M1 macrophage polarization, ↑Arg-1, ↑iNOS	50 mg/kg of mice, i.p.	[187]
	Synthetic	JEV-infected model	↓ iNOS, ↓ TNF-α, ↓ IFN-γ, ↓ MCP-1, ↓ IL-6, ↓ p-p38 MAPK, ↓ p-c-Jun ↓ p-ERK-1/2, ↑ p-Akt	10 mg/kg of mice, i.p.	[152]
		LPS-injured model	↓ nitrate/nitrite ratio, ↓ iNOS, ↓ TNF-α, ↓ IL-6, ↓ MIP-2, ↓ p-ERK1/2, ↓ p-JNK, ↓ p-p38	50 mg/kg of mice, i.p.	[153]
		EAE model	↓IFN-γ, ↓T-bet, ↓IL-17, ↓ROR-γt, ↓Th1, ↓Th17	5–10 mg/kg of mice, i.p.	[217]
		ConA-induced acute hepatitis	↑IL-4, ↓F4/80, ↓CD49b, ↓CD4 T cells	5–10 mg/kg of mice, i.p.	[188]
		AMI model	↓ iNOS, ↓ COX-2, ↓ IL-1β, ↓ IL-6, ↓ p-ERK1/2	100–200 μmol/kg of rat	[155]
		BLM-induced skin fibrosis model	↓ TGF-β1, ↓ IL-1β, ↓ IL-4, ↓ IL-6, ↓ TNF-α, ↓ MCP-1	3 mg/kg of mice, i.p.	[156]
		DMM model	↓cartilage erosion, ↓hypocellularity, ↓proteoglycan loss	30 mg/kg of mice, p.o.	[189]
		imiquimod-induced murine psoriasis model	↑p-CREB, ↑cAMP, ↑IL-10, ↓TNF-α, IFN-γ, ↓COX-2, ↓iNOS, ↓IL-2, ↓IL-6, ↓IL-12, ↓IL-17, ↓IL-22, ↓IL-23, ↓IL-27	5% cream	[190]
		silicosis model	↓TGF-β, ↓TLR-4	30–60 mg/kg of rat, i.g.	[88]
		cadmium-intoxicated model	↓NF-κB p65, ↓TNF-α, ↓IL-1β	80 mg/kg of rat, i.g.	[157]
Hinokinin	Synthetic	high-fat diet/STZ-induced type 2 diabetic	↓TLR 4, ↓MYD88, ↓NF-κB p65, ↑IKBα, ↓TNF-α, ↓IL-1β, ↓p38, ↓ERK 1/2, ↓JNK, ↓MEK	20–40 mg/kg of mice, p.o.	[158]
Matairesinol	Synthetic	IRBP/CFA-induced EAU model	↓T17 cells, ↓IL-17A, ↓IL-17F, ↓IL-21, ↓GM-CSF, ↓IRF-4, ↓Hif1, ↓Batf, ↓ROR-γt, ↓TNF-α	1 mg/kg of mice, i.p.	[192]
		CLP-induced sepsis	↓TNF-α, ↓IL-1β, ↓IL-6, ↓IFN-γ, ↓IL- 8, ↓MCP1, ↓MAPK, ↓JNK, ↓NF-κB	5–20 mg/kg of rat, p.o.	[93]
Nortrachelogenin	Synthetic	carrageenan-induced paw edema	↓paw edema volume	100 mg/kg of mice, i.p.	[194]
**Furanoid structure group**					
Nectandrin B	*Guaiacum officinale* L.	IL-1β-treated rat hepatocytes	↓ NO level	IC_50_ 43.4 μM	[218]
Taxiresinol	*Taxus baccata* Linn.	carrageenan-induced paw edema	↓paw edema volume	100 mg/kg of mice, p.o.	[219]
**Furofuranoid structure group**					
Acanthoside B	*Salicornia europaea* Linn.	Amnesic AD-like model	↓iNOS, ↓COX-2, ↓TNF-α, ↓IL-1β, ↓IL-6, ↑IL-10	10, 20 mg/kg of mice, p.o.	[220]
(+)-Diayangambin	*Piper fimbriulatum*	carrageenan-induced paw edema	↓paw volume, ↓prostaglandin E2	40 mg/kg of mice, p.o.	[197]
Fargesin	*Magnolia* sp.	DSS-induced colitis	↓inflammatory infiltration, ↓MPO, ↓TNF-α, ↓NO, ↑IκBα, ↓NF-κB	50 mg/kg of mice, p.o.	[199]
	Synthetic	ApoE^−/−^ model	↓macrophage infiltration, ↑M2 phenotype polarization, ↓TNF-α, ↓IL-1β, ↓IL-6, ↓MCP-1, ↑IL-10	50 mg/kg of mice, p.o.	[221]
Isoeucommin A	*Eucommia ulmoides* Oliv.	STZ-induced diabetic nephropathy	↓immune infiltration, ↓TNF-α, ↓IL-1β, ↓IL-6	2.5–10 mg/kg of rat, i.v.	[104]
Koreanaside A	*Forsythia koreana*	DSS-induced acute colitis	↓iNOS, ↓COX-2, ↓IL-6, ↓TNF-α ↓p-c-Fos, ↓p-p65, ↓p-STAT1, ↓p-STAT3	5–20 mg/kg of mice, i.p.	[200]
Phillygenin	*Forsythia koreana*	carrageenan-induced paw edema	↓paw volume	12.5–100 mg/kg of mice, i.p.	[201]
	*Forsythia fructus*	CCl_4_-induced liver fibrosis	↓LPS, ↓MIP-1, ↓TNF-α, ↓IL-1β, ↓IL-6, ↓immune infiltration	20, 40 mg/kg of mice, i.g.	[222]
Pinoresinol diglucoside	Synthetic	Aβ-infused model	↓TLR4, ↓NF-κB p65, ↓TNF-α, ↓IL-1β	5–10 mg/kg of mice, i.g.	[160]
		MCAO model	↓TNF-α, ↓IL-1β, ↓IL-6, ↓p-IKKβ, ↓p-IkBα, ↑cNF-κB p65, ↓p-p65	5–10 mg/kg of mice, i.v.	[161]
Sesamin	*Sesamum indicum* Linn.	fMLF-induced inflammation in a murine air-pouch model	↓leukocyte infiltration	12 mg/kg of mice, i.p.	[203]
		PCA model	↓PCA reaction	50–200 mg/kg of rat, p.o.	[204]
		LPS-treated model	↓TNF-α, ↓MCP-1, ↓IL-1β	10 mg/kg of rat, p.o.	[168]
		LPS-treated model	↓NF-κB, ↓TLR4, ↓Cox2, ↓TNF-α, ↓IL-6	100 mg/kg of mice, p.o.	[169]
		DSS-induced colitis model	↓IL-6, ↓IL-1β, ↓TNF-α	50–100 mg/kg of mice, i.g.	[113]
		cisplatin-injured model	↓TNF-α, ↓IL-1β, ↓TGF-β1, ↓MPO	5 mg/kg of rat, p.o.	[170]
	Synthetic	CCl_4_-induced hepatotoxicity model	↓TNF-α	60–120 mg/kg of mice, p.o.	[164]
		carrageenan-induced lung inflammation	↑A20, ↑TAX1BP1, ↓IL-6, ↓IL-8, ↓IL-1β, ↓TNF-α, ↓MIP-2, ↓MPO, ↓β-glucuronidase, ↓p-p65, ↓TRAF6	50–100 mg/kg of rat, p.o.	[205]
		fluoride-exposed model	↓TNF-α	0.5–1 g/kg of carp, diet	[165]
Syringaresinol	*Rubia philippinensis*	carrageenan-induced paw edema model	↓paw edema volume	50 mg/kg of mice, p.o.	[207]
		CLP-induced sepsis	↓TNF-α, ↓IL-6, ↓IL-18, ↓IL-1β	50 mg/kg of mice, p.o.	[208]
		STZ-induced type 1 diabetic model	↓macrophage, monocyte, neutrophil infiltration in myocardium, ↓TNF-α, ↓IL-6, ↓IL-1β	25 mg/kg of mice, p.o.	[116]
	*Albiziae cortex*	LPS-treated model	↓IL-6, ↓IL-1β, ↓TNF-α, ↓COX-2, ↓iNOS, ↓microglia activation	60 mg/kg of mice, p.o.	[209]

4-HNE, 4-hydroxy-2-nonenal; AD, Alzheimer’s disease; Ang, angiotensin; ARDS, acute respiratory distress syndrome; BALF, brochoalveolar lavage fluid; Batf, basic leucine zipper transcriptional factor ATF-like; CFA, Complete Freund’s Adjuvant; CLP, cecal ligation and puncture; ConA, concanavalin A; CRP, C-reactive protein; DTH, delayed type hypersensitivity; DMM, destabilization of the medial meniscus; DNFB, 2,4-dinitro-1-fluorbenzene; DSS, dextran sulfate sodium; EAE, experimental autoimmune encephalomyelitis; EAU, experimental autoimmune uveitis; fMLF, N-Formyl-Met-Leu-Phe; G-003M, formulation of synthetic podophyllotoxin with rutin; GM-CSF, granulocyte-macrophage colony-stimulating factor; HCC, hepatocellular carcinoma; Hif1, hypoxia-inducible factor-1; i.a., intra-articular injection; i.g., intragastric administration; i.m., intramuscular injection; i.n., intranasal administration; i.p., intraperitoneal injection; i.v., intravenous injection; IRBP, inter photoreceptor binding protein; IRF-4/8, interferon regulatory factor 4/8; JEV, Japanese encephalitis virus; MCAO, middle cerebral artery occlusion model; MDSCs-G/M, myeloid-derived suppressor cells granulocytic/monocytic; MIP, macrophage inflammatory protein; OVA, ovalbumin; PAA, polyamidoamine; PC, picryl chloride; PCA, passive cutaneous anaphylaxis; PGN, peptidoglycan; p.o., oral administration; RCA, reversed cutaneous anaphylaxis; ROR-γt, retineic-acid-receptor-related orphan nuclear receptor gamma; SRBC, sheep red blood cell; STZ, streptozotocin; TGR, thoracic gamma radiation; Th2, T-helper 2; TLR4, Toll-like receptor 4; TNBS, 2,4,6-trinitrobenzene sulfonic acid; TPA, 12-O-tetradecanoylphorbol-13-acetate; XOD, xanthine oxidase. Downward-pointing red arrows reflect the downregulatory action, upward-pointing green arrows reflect the upregulatory action.

### 3.5. Dibenzylbutyrolactone and Dibenzylbutyrolactol Skeletons

#### 3.5.1. Arctigenin

Arctigenin showed strong DPPH radical scavenging activity in vitro and in vivo, improved survival of *C. elegans* under oxidative stress [84]. Arctigenin had multiple effects in various cellular and animal models, inhibiting ROS-induced activation of ERK1/2, PI3K/Akt, IKKβ, and NF-κB signaling pathways, followed by the suppression of pro-inflammatory enzymes such as COX, LOX, phospholipase A2 (PLA2) and phosphodiesterase (PDE), decrease in production of TGF-β1 and inflammatory cytokines (IL-1β, IL-4, IL-6, TNF-α, MCP-1). At the same time, additional activation of the AMPK and Nrf2 signaling pathways by arctigenin increased the expression of genes associated with antioxidants, including SOD, GR, GPx, Trx, HO-1, CAT, uncoupling protein 2 (UCP2), and promoted the secretion of IL-10 [80,83,155,156,157,186,216]. Arctigenin also inhibited ROS production in H_2_O_2_-treated rat primary astrocytes [82], scavenged free radicals and reduced the level of cellular peroxide in glutamate-injured rat cortical neurons by directly binding to kainic acid receptors [77], thereby demonstrating its neuroprotective potential.

Arctigenin protected against LPS-induced lung inflammatory and oxidative damage [153], and, in IL-1β–induced human osteoarthritis (OA) chondrocytes and mouse OA model, the lignan effectively decreased the level of pro-inflammatory mediators attenuating the progression of OA [189]. Arctigenin inhibited allergic inflammation type I as heterologous passive cutaneous anaphylaxis, type II as reversed cutaneous anaphylaxis, type III as the sheep red blood cell-induced Arthus reaction and contact dermatitis (type IV hypersensitive inflammation) in vivo as well as suppressed pro-inflammatory enzymes, such as COX, LOX, phospholipase A2 (PLA_2_), and phosphodiesterase (PDE), in vitro [186]. Arctigenin possessed anti-inflammatory and immunosuppressive properties by inhibiting Th17 cell differentiation and proliferation in a model of experimental autoimmune encephalomyelitis (EAE), which indicates onits therapeutic potential for multiple sclerosis treatment [217].

In another study, arctigenin showed anti-inflammatory effects in peptidoglycan- or LPS-induced peritoneal macrophages in vitro, LPS-induced systemic inflammation in vivo, and 2,4,6-trinitrobenzene sulfonic acid-induced colitis model [216]. Arctigenin also ameliorated LPS-induced inflammation in vitro and in vivo by enhancing the accumulation of granulocytic myeloid-derived suppressor cells (G-MDSCs) through miR-127-5p/IRF8 axis and the immunosuppressive role of MDSCs through the upregulation of Arg-1 and iNOS, thereby protecting from acute lung injury [187]. Arctigenin inhibited pro-inflammatory cytokines and chemokines in LPS-induced RAW264.7 murine macrophage-like cells in vitro as well as in vivo, remarkably reduced the congestion and necroinflammation of liver, decreased the infiltration of CD4 T and NKT cells and macrophages into the liver and suppressed T lymphocyte proliferation in a murine model of concanavalin A-induced acute hepatitis [188]. In LPS-primed human PBMCs and murine RAW264.7 cells in vitro and in imiquimod-induced murine psoriasis model in vivo, arctigenin inhibited activity of PDE4 and activated the cAMP-dependent protein kinase A/cAMP-response element binding protein (PKA/CREB) signaling, ameliorated psoriatic manifestations by decreasing the adhesion and chemotaxis of inflammatory cells, rectifying the immune dysfunction and hyperactivation of keratinocytes in the inflamed skin microenvironments, reducing the production of inflammatory cytokines and promoting the secretion of IL-10 [78,190].

In mice infected by Japanese encephalitis virus, arctigenin demonstrated a marked decrease in the levels of stress-associated signalling molecules, ROS/RNS and pro-inflammatory cytokine production, thereby reducing neuronal death, secondary inflammation and oxidative stress resulting from microglial activation [152]. Arctigenin protected against TGF-β1-induced upregulation of a key mediator of tubulointerstitial inflammation MCP-1 and the resulting epithelial–mesenchymal transition-like phenotypic changes, thereby declaring arctigenin as therapeutic agent to treat renal tubulointerstitial fibrosis [83]. In bleomycin-induced skin fibrosis murine model, arctigenin reduced inflammation and oxidative stress, inhibited the transformation of fibroblasts into myofibroblasts, also showing antifibrotic potential [156]. In addition, arctigenin ameliorated silicosis-associated oxidative stress, immune-related inflammatory reaction, and fibrosis both in vitro and in vivo by inhibiting TLR-4/NLRP3/TGF-β signal transduction and increasing the mitochondrial membrane potential, which in turn inhibited the production of ROS, the polarization of macrophages, and the differentiation of myofibroblasts [88].

Arctigenin demonstrated anti-ulcer activity by reducing oxidative and inflammatory damage in the ethanol- and acetic acid-induced ulcerogenic models viathe decrease in the levels of MDA, TNF-α, IL-6, IL-10, and C-reactive protein and the increase in the level of SOD in serum [151]. Arctigenin showed antiarrhythmic protective effect via the attenuation of myocardial ischemia/reperfusion (MI/R) injury by increasing the activities of antioxidant enzymes and reducing the level of MDA [154]. Arctigenin also attenuated apoptosis, inflammation, and oxidative stress in oxygen glucose deprivation-treated cardiomyocytes, improved the heart functions and decreased the infarct size in the acute MI/R-rats [87,155]. On the normal WRL68 hepatocytes exposed to oleic acid accumulation, arctigenin demonstrated a protective effect on cell survival, lipid metabolism, oxidation stress, and inflammation [85]. Arctigenin mitigated cadmium-induced nephrotoxicity in rats by increasing GSH level and antioxidant enzyme activity providing protection against oxidative DNA damage [157].

A number of studies have suggested arctigenin as a potential chemotherapeutic agent against various premalignant and malignant cells. For example, arctigenin was shown to promote glucose-starved tumor cells to undergo necrosis by elevating the iROS level and inhibiting mitochondrial respiration and cellular energy metabolism in general [79]. Arctigenin also induced apoptosis of human breast cancer MDA-MB-231 cells via the triggering of the mitochondrial caspase-independent ROS/p38 MAPK pathway and epigenetic Bcl-2 downregulation [81]. In another study, arctigenin promoted apoptosis in human hepatocellular carcinoma-related Hep G2 cells via the enhancement of theROS-mediated mitochondrial dysfunction, p38 and JNK and MAPK pathways activation as well as CYP450 and Bax upregulation [86].

The potential toxicity of arctigenin was demonstrated in a toxicological study of the subchronic toxicity profile for 28 days, where even the lowest dose of 12 mg/kg caused significant side effects in rats, including accumulation of arctigenin in organs and irreversible side effects in several tissues (focal necrosis, lymphocytic infiltration in the renal cortex, liver lobules and prostate) [223].

#### 3.5.2. Carissanol

Carissanol showed moderate DPPH free radical scavenging activity three times weaker than a well-known antioxidant quercetin [96], although in another study (–)-carissanol showed almost equal potential like that of Trolox [224].

#### 3.5.3. Hinokinin

Hinokinin exerted a significant antioxidant effect by inhibiting the accumulation of H_2_O_2_ produced by *Trypanosoma cruzi* mitochondria in the presence of the pro-oxidant compound, tert-butyl hydroperoxide, showed a protective effect against the chromosome damage induced by the free radicals generated by doxorubicin [89]. In another study, hinokinin inhibited the secretion of TNF-α and IL-6 in LPS-stimulated human leukemia monocytic (THP-1) cell line [191]. In the high fat diet/streptozotocin-induced cardiac injury model of type 2 diabetes, hinokinin significantly protected against oxidative stress, inflammation, and apoptosis via the modulation of Nrf2/Keap1/ARE pathway, MAPKs (JNK, p38 and ERK1/2) and TLR4/MyD88/NF-κB mediated inflammatory pathways and mitochondrial-dependent (intrinsic) apoptosis pathway [158].

#### 3.5.4. Matairesinol and Its Derivatives

Matairesinol showed strong superoxide, peroxyl, and DPPH radical scavenging activity and improved survival of *C. elegans* under oxidative stress [69,70,84,90,91]. Matairesinol suppressed mitochondrial ROS generation and decreased hypoxia-inducible factor-1α (HIF-1α) in hypoxic HeLa cells, inhibited proliferation of human umbilical vein endothelial cells [92]. In another study, matairesinol in vitro inhibited Th17 cell differentiation via MAPK/ROR-γt signaling pathway and in vivo restrained an interphotoreceptor retinoid-binding protein-specific Th17 proliferation, infiltration and cytokine production, alleviated intraocular inflammation in the eye in the model of experimental autoimmune uveitis [192]. In rat sepsis model by CLP and the model of LPS-induced neuronal damage, matairesinol exerted neuronal protection, anti-inflammatory and anti-oxidative stress effects by upregulating AMPK and Nrf2/HO-1 pathways and inactivating the MAPK and NF-κB pathways, thereby ameliorating sepsis-mediated brain injury [93].

7′-hydroxymatairesinol showed antioxidative potency and peroxyl radical scavenging capacity similar to that of Trolox [69] and also exhibited excellent scavenging of DPPH radicals reacting at the level of ascorbic acid and surpassing 2,6-di-tert-butyl-4-methylphenol (BHP) in activity [70,94]. In addition, 7′-hydroxymatairesinol demonstrated significant anti-inflammatory effects by inhibiting NF-κB and ERK phosphorylation in human aortic endothelial cells [193].

#### 3.5.5. Nortrachelogenin

(–)-Nortrachelogenin showed strong antioxidative potency and DPPH free radical, superoxide, and peroxyl radical scavenging activity, more effective than both BHA and Trolox [56,69,70,90,96,97]. Nortrachelogenin was also able to reduce the basal level of peroxides in Jurkat cells as well as counteracted peroxide increase induced by BHP treatment [97]. In another study, (+)-nortrachelogenin showed moderate anti-DPPH free radical activity 4.5 times weaker than a well-known antioxidant L-ascorbic acid [95]. The anti-inflammatory effect of nortrachelogenin was demonstrated in amouse carrageenan-induced paw inflammation model and on LPS-activated murine J774 macrophages [194].

### 3.6. Furanoid Skeletons

#### 3.6.1. Lariciresinol

(+)-Lariciresinol possessed significant antioxidant potential in various in vitro models. Lariciresinol revealed very strong DPPH-, ABTS-, superoxide-, and hydroxyl-radical scavenging activity, exhibited strong lipid peroxidation inhibition as well as ferric- and cupric-reducing power [26,27,69,70,98,99]. Lariciresinol inhibited ROS generation in HL-60, AAPH-treated and intact RAW 264.7 cells. Also, it increased transcriptional and translational levels of detoxifying antioxidant enzymes via Nrf2-p38 signaling [26,99].

#### 3.6.2. Nectandrin B

Nectandrin B reduced senescence of cells by reducing cellular oxidative stress via the direct radical scavenging or indirectly via the induction of antioxidant enzymes through the activation of AMPK. Nectandrin B exerted DPPH radical scavenging equal to the well-known antioxidants Tempo and NAC, significantly reduced H_2_O_2_- and palmitic acid-induced iROS production in both young and old human diploid fibroblasts (HDFs), stimulated the expression of SOD I and II in old HDFs, increased activation of PI3K and Akt, and reversed the activity of ERK1/2 and p38 [100]. In another study, nectandrin B showed an anti-inflammatory effect by inhibiting NO production in IL-1β-stimulated rat hepatocytes [218].

#### 3.6.3. Olivil

(–)-Olivil showed moderate anti-DPPH free radical activity [56] and also possessed moderate anti-NO production activity in LPS-activated RAW264.7 cells [195].

#### 3.6.4. Taxiresinol

Taxiresinol possessed strong DPPH radical scavenging activity (IC_50_ 18.4 μM), superior to that of a well-known antioxidant caffeic acid (IC_50_ 25.5 μM), and comparable to the activity of ascorbic acid (IC_50_ 12.6 μM) [68]. Taxiresinol exhibited strong anti-inflammatory potential by inhibiting the release of leukotriene C4 (LTC_4_) in rat basophilic leukemia (RBL-1) cells [196], showed moderate inhibitory activity of NO production in LPS-activated RAW264.7 cells [195]. Taxiresinol weakened the carrageenan-induced inflammatory process by reducing the swelling thickness of the hind paw by 27% [219].

### 3.7. Furofuran Skeletons

#### 3.7.1. Acanthoside B

Acanthoside B isolated from *Salicornia europaea* substantially attenuated AD-like amnesic traits in mice by restoring the cholinergic activity, endogenous antioxidant status, and suppressing neuroinflammation [220].

#### 3.7.2. Dendranlignan A

A bisepoxylignan dendranlignan A, isolated from the flowers of *Dendranthema morifolium* (Ramat.), was found to inhibit the ROS formation and decrease the levels of in- flammatory cytokines and the nuclear localization of c-JUN, p65 and IRF3 in LPS-induced H9c2 cells, thereby suggesting that the lignan inhibits TLR4 signaling. Although no effect on TLR4 production was shown, it was predicted by molecular docking that dendranlignan A can occupy the ligand-binding sites of TLR4 receptor [103].

#### 3.7.3. Diayangambin

(+)-Diayangambin showed immunosuppressive and anti-inflammatory activity in vitro and in vivo. Diayangambin inhibited human mononuclear cell proliferation and reduced the level of PGE_2_ in stimulated RAW 264.7 macrophages. When administered orally, diayangambin reduced ear swelling in mice treated by 2,4-dinitrofluorobenzene. Also, in a model of carrageenan paw edema, this lignan significantly decreased inflamed paw volume and PGE_2_ levels [197].

#### 3.7.4. Fargesin

Fargesin was found in several species of Magnolia. It demonstrated anti-inflammatory properties via the suppression of PKC pathway, including downstream JNK, nuclear factors AP-1, and NF-ĸB [198]. In another study, fargesin ameliorated chemically induced inflammatory bowel disease in mice, reducing inflammatory infiltration, production of NO and cytokines. The effects were shown to be associated with NF-κB signaling suppression [199]. Fargesin also inhibited atherosclerosis in experimental mice by reducing inflammatory response via TLR4/NF-κB pathway [221]. In a rat MI/R injury model, it was shown that fargesin also had antioxidant properties by reducing the level of ROS and MDA and increasing the level of antioxidant enzymes [159].

#### 3.7.5. Isoeucommin A

Isoeucommin A, a lignan compound from *Eucommia ulmoides* Oliv, showed protective effects in diabetic nephropathy and alleviated kidney injury by reducing inflammation and oxidative stress in vitro and in vivo [104].

#### 3.7.6. Koreanaside A

Koreanaside A showed high radical scavenging activity with oxygen radical absorbance capacity (ORAC) values of 0.97 ± 0.01 and inhibited TNF-α-induced vascular cell adhesion molecule-1 (VCAM-1) expression in mouse vascular smooth muscle (MOVAS) cells [105]. Koreanaside A inhibited the activation of activator protein-1 (AP-1), NF-κB, and JAK/STAT pathways and the subsequent induction of pro-inflammatory mediators in LPS-stimulated RAW 264.7 macrophages and dextran sulfate sodium (DSS)-induced colitis mice models, thus demonstrating a potential in the treatment of inflammatory bowel disease [200].

#### 3.7.7. Phylligenin

Phylligenin demonstrated anti-inflammatory activity in vitro and in vivo by reducing the production of PGE_2_ and NO, suppressing NK-κB activation, and inhibiting carrageenan-induced paw edema in mice. Phylligenin also attenuated liver fibrosis partly via the modulation of inflammation and gut microbiota [201,222].

#### 3.7.8. Pinoresinol and Its Derivatives

Pinoresinol exhibited a significant antioxidant potential in hydroxyl and DPPH radical scavenging assays [27,56,70,97] with ORAC comparable to Trolox [105]. Pinoresinol showed moderate inhibitory activity against lipid peroxidation induced by non-enzymic Fe(II)-ascorbic acid system in rat liver microsomes [107] and 1.8 times stronger inhibitory activity to Cu^2+^-induced low-density lipoprotein (LDL) oxidation over probucol [106]. In intact human lung epithelial Beas-2B cells, pinoresinol promoted the nuclear translocation and stabilization of Nrf2 followed by the activation of downstream cytoprotective genes, NQO1 and γ-GCS. Futhermore, pinoresinol treatment protected human lung epithelial cells against sodium arsenite-induced oxidative insults by increasing the intracellular GSH level and inhibiting ROS production via Nrf2-mediated antioxidant response [108]. Pinoresinol also demonstrated anti-inflammatory properties by decreasing the secretion of PGE_2_, IL-6, and MCP-1 (but not IL-8) and inhibiting the NF-κB activation in IL-1β-treated human colon adenocarcinoma (Caco-2) cells [202].

4-Ketopinoresinol has been shown to possess a significant antioxidant potential. 4-Ketopinoresinol displayed strong DPPH free radical scavenging activity and the ability to reduce the basal level of peroxides in Jurkat cells, counteracted BHP-induced peroxide increase [97,101]. 4-Ketopinoresinol protected against H_2_O_2_-induced cell injury, oxidative stress-induced DNA damage and cell death via scavenging ROS directly and increasing Akt phosphorylation with the following nuclear translocation of Nrf2 for the expression of ARE-dependent cytoprotective genes [102].

Pinoresinol diglucoside (PDG) inhibited oxidized low-density lipoprotein, (oxLDL)-induced upregulation of the ROS production and lipid peroxidation, relieved the inhibition of SOD activity, and inhibited p38MAPK/NF-κB activation in human umbilical vein endothelia cells (HUVECs) [109]. Intragastric administration of PDG ameliorated memory dysfunction and attenuated neuroinflammation, neuronal apoptosis and oxidative stress through the TLR4/NF-κB and Nrf2/HO-1 pathways in the model of Aβ-infused mice. Mechanistically, PDG restrained the release of pro-inflammatory cytokines (TNF-α and IL-1β), ROS, and MDA, promoted the activity of the antioxidant enzymes (SOD and CAT), Nrf2 and HO-1 expression, upregulated the ratio of Bcl-2/Bax and downregulated cytochrome C as well as significantly reduced the expression of TLR4 and the activation of NF-κB p65 [160]. PDG also alleviated inflammation and oxidative stress developing during the MCAO-induced neurological dysfunction of the mice, reducing the infarct volume, brain water content, and neuron injury [161].

#### 3.7.9. Sesamin

Sesamin, an abundant lignan in sesame seeds and oil, ameliorated oxLDL-mediated vascular endothelial dysfunction and showed the antiatherogenic action through its ability to counteract the ROS generation, the impairment of antioxidant enzymes, and the activation of the expression of IL-8 and NF-κB [110]. As an important component of the immune reaction, the response of human leukocytes to chemoattractants can also be regulated by sesamin. Sesamin significantly attenuated bacterial chemotactic peptide fMLF-induced leukocyte chemotaxis and inhibited ERK1/2 phosphorylation and NF-κB activation in ETFR cells expressing fMLF-specific receptor FPR in vitro and, in a murine air-pouch model in vivo, suppressed leukocyte infiltration induced by fMLF [203].

In addition, sesamin attenuated allergic responses mediated by mast cells in vivo and in vitro. Sesamin inhibited IgE-induced anaphylactic reactions by blocking histamine release and pro-inflammatory cytokine expression [204]. In aortic tissue of diabetic rats, sesamin attenuated oxidative stress by reversing the increased MDA content and the reduced activity of SOD [162]. Sesamin protected the femoral head from osteonecrosis by inhibiting the ROS-induced osteoblast apoptosis [112]. Sesamin protected against ulcerative colitis in colorectal adenocarcinoma Caco-2 cells injured by H_2_O_2_-induced oxidative stress [113]. Sesamin feeding protected adult *Drosophila* against oxidative damage [171].

The neuroprotective potential of sesamin has been demonstrated in several studies. Sesamin showed reversal effect in unilateral striatal 6-hydroxydopamine (6-OHDA) model of Parkinson’s disease (PD), which included attenuation of oxidative stress via lowering striatal level of MDA and ROS and improving SOD activity [167]. In the kainic acid-induced status epilepticus brain injury model, sesamin demonstrated anti-inflammatory and antioxidative effects via decreasing MDA content and ROS level as well as reducing PGE_2_ production [111]. Sesamin also demonstrated neuroprotective effects in H_2_O_2_-treated human neuroblastoma (SH-SY5Y) through the expression of the antioxidant enzymes leading to the elimination of the excessive ROS production, activation of SIRT1-SIRT3-FOXO3a expression and upregulation of anti-apoptotic Bcl-2 [114].

Sesamin decreased oxidative stress and injury in the liver and kidneys of doxorubicin-treated rats [166]. Sesamin significantly prevented nickel-induced hepatotoxicity. Sesamin reversed the elevation of ROS production and depletion of the intracellular GSH level, restored the activities of antioxidant enzymes and decreased 8-OHdG levels (marker of oxidative DNA damage), increased expression levels of PI3K and phosphorylated Akt, which in turn led to the inactivation of pro-apoptotic signaling events in the liver of nickel-treated mice [163]. Sesamin also mitigated CCl_4_-induced hepatotoxicity by suppressing the elevation of the ROS production, oxidative stress, and apoptosis in mouse liver [164]. Orally supplemented sesamin acted as a protective agent against LPS-induced lipid peroxidation in both serum and liver by restoring the loss of SOD (but not CAT and GR) activity and reducing serum levels of TNF-α, MCP-1, and IL-1β [168]. In another study, sesamin attenuated carrageenan-induced lung inflammation and injury in rats by indirectly inhibiting NF-κB pathway through the upregulation of A20 and TAX1BP1 [205]. It also abrogated LPS-induced acute kidney injury via the attenuation of renal oxidative stress, inflammation, and apoptosis, and returning the renal oxidative stress-related parameters [169], thereby demonstrating potential against injuries in septic conditions. Sesamin possessed mitigative action on cisplatin (CP) nephrotoxicity in rats by reversing the CP-induced oxidative stress and inflammation [170]. In addition, sesamin significantly alleviated fluoride-induced renal oxidative stress and apoptosis of carp *Cyprinus carpio* [165].

*Epi*-sesamin induced potent inhibition of endothelial protein C receptor shedding induced by PMA, TNF-α, IL-1b, and CLP operation, which affected the regulation of blood coagulation [206].

#### 3.7.10. Syringaresinol

Syringaresinol (also known as lirioresinol B) exhibited potent antioxidant activities in hydroxyl and DPPH radical scavenging assays [27,101]. In a cardiomyocyte model of I/R injury following H/R, syringaresinol decreased the levels of ROS, increased the expression of antioxidant genes, and also stimulated the nuclear localization and activity of FOXO3 followed by the degradation of HIF-1α that provided a protective effect against cellular damage and death [115]. Syringaresinol oral administration lowered the levels of lipid peroxide marker 8-isoprostane and iROS in cultured primary fibroblasts from the skin of Sod1^–/–^ mice as well as regulated the FOXO3/MMP-2 axis in oxidative-damaged skin and exhibited beneficial effects on age-related skin involution [172].

Anti-infammatory efficacy of syringaresinol was demonstrated in various in vitro and in vivo experiments. In LPS-stimulated RAW 264.7 cells and in a carrageenan-induced hind paw edema assay, syringaresinol downregulated NF-κB expression by interfering with JNK and p38 phosphorylation followed by the decrease in mRNA levels of iNOS, COX-2, TNF-α, IL-1β, and IL-6, thus suggesting its significant therapeutic potential [207]. Syringaresinol inhibited the release of LTC_4_ in RBL-1 cells [196].

Syringaresinol also played a protective role against sepsis-induced cardiac inflammation and LPS-induced cardiomyocyte damage, suggesting its possible role in alleviating cardiac dysfunction. Levels of inflammatory cytokines were significantly reversed by the administration of the lignan after the increase elicited by CLP operation. The activation of estrogen receptor was shown to be essential for the cardioprotective function of syringaresinol [208]. On streptozotocin-induced type 1 diabetic mice and high glucose-injured neonatal cardiomyocytes, syringaresinol demonstrated anti-inflammatory, anti-fibrotic, and anti-oxidant effects through Keap1/Nrf2 system and TGF-β/Smad signaling pathway [116]. Thus, the above series of studies allows syringaresinol to be considered as a therapeutic agent for the treatment of cardiomyopathy. In addition, syringaresinol demonstrated anti-neuroinflammatory effects in microglia cells and wild-type mice. Estrogen receptor β was found to be implied in the anti-inflammatory activity of the lignan in BV2 microglia [209].

## 4. Main Approaches for Lignan Synthesis

Reviews over the past decades describe a huge variety of different approaches for the racemic or enantioselective synthesis of plant lignans. Overall, the key step is the dimerization of two monomers that are derivatives of coniferyl alcohol. The difference among chemical strategies for obtaining lignans depends on the skeleton peculiaritiesprovided by natural biosynthesis. Wide varieties of elegant methods to obtain these compounds have been already published. Classical pathways include oxidative dimerization, the Diels-Alder reaction, aldol reaction, and the Stobbe condensation [225,226].

### 4.1. Oxidative Dimerization

The first attempts to synthesize lignans by oxidative dimerization of cinnamon derivatives included the oxidative coupling of phenols and their esters in the presence of various enzymes. Thus, synthetic analogs of pinoresinol and syringaresinol, respectively, were obtained from cinnamic alcohol derivatives under the catalytic action of enzymes isolated from *Caldariomyces fumago* [227] and other fungi [228], respectively (Figure 1). The final compounds were obtained as a racemic mixture of isomers, and there was an overall low yield due to the side products formation.

The method of oxidative dimerization of cinnamic acid esters derivatives with the use of inorganic compounds as catalysts emerged as the most effective from a practical point of view. The use of oxidizing agents such as FeCl_3_, K_3_[Fe(CN)_6_], Ag_2_O, MnO_2_, H_2_O_2_, Pb(C_2_H_3_O_2_)_4_ and various peroxidases is common in the developing of synthetic methods for obtaining analogs of natural lignans [225].

Nowadays, a large number of papers are devoted to investigation of an oxidizing agent effect on the oxidative coupling reaction efficiency. Thus, in the work of Maeda et al., their influence on dimerization of methyl (E)-3-(4,5-dihydroxy-2-methoxyphenyl) propionate was studied [229]. By-products predominantly were formed when Ag_2_O was used in a benzene–acetone mixture at room temperature, and the yield of the target dihydronaphthalene derivative was quite low (28%). The obtained compounds were subjected to acylation reaction in the presence of pyridine for a more accurate separation by column chromatography. Figure 2 describes in more detail possible by-products of the oxidative dimerization carried out in the presence of Ag_2_O.

Further attempts to use this method for phenylpropanoid dimerization confirmed negative role of Ag_2_O leading to the formation of a complex mixture that makes it difficult to separate the target product. Thus, attempt of dimerization in methylene chloride with two equivalents of silver oxide at room temperature led to the predominant formation of neolignans instead of lignans. The yield of 1,2-dihydronaphthalene derivative was less than 1%. The shift of this reaction towards neolignans formation was also confirmed by Lemierre et al. [230].

In the same work, it was found that the presence of Mn^2+^ ions shift the dimerization towards the formation of benzoxanthine derivatives. Thus, Dakino et al. performed the oxidative coupling reaction of caffeic acid phenethyl ester in methylene chloride in the presence of 10 equivalents of MnO_2_ for 4 h at room temperature. Further analysis of the reaction mixture by HPLC-UV revealed two main products: dihydronaphthalene derivative and benzoxanthine (Figure 3). After purification on silica gel using column chromatography, the total yield of 16.5% for the first compound and 48% for the second one was achieved. When the solvent was changed to chloroform, the ratio of the obtained substances changed in favor to the benzoxanthine derivative (51%), while only 7% of the aryl dihydronaphthalene lignan was formed. The use of Mn(acac)_2_ instead of MnO_2_ led to a similar result, and dihydrobenzofuran neolignans were not observed in the reaction mixture. To shift the reaction direction towards benzoxanthine lignans, 4 equivalents of Mn(OAc)_3_ in chloroform were used. A larger amount of oxidizing agent led to tarring of the reaction mixture and the yield reduction. As a result, 71% of benzoxanthine lignan and 22% of aryldihydronaphthalene derivative were formed (Figure 3) [231].

In 2015, a synthesis of three arylnaphthalene derivatives (retrojusticidin B, justicidin E, and helloxanthin) was published using Mn(OAc)_3_ as an oxidizing reagent [232]. In this publication, dimerization of carbonitrile was obtained by Knoevenagel condensation of α-cyanoether and an aldehyde with the yield of 67%. Next, the intermediate obtained as a result of dimerization was reduced to the final lignan (Figure 4).

An aqueous solution of 0.1 M KMnO_4_ was also tested as an oxidizing agent. In this case, the oxidative coupling was performed from caffeic acid in distilled water in the presence of 0.2 equivalents of potassium permanganate. A 1,2-dihydronaphthalene compound was formed as a by-product in the reaction with the loss of one carboxyl group. The yields observed in this reaction were quite low [233] (Figure 5).

In Maeda’s study the selection of potassium hexacyanoferrate K_3_[Fe(CN)_6_] as an oxidizing agent was also studied. Oxidative coupling in the presence of 1.2 equivalents of K_3_[Fe(CN)_6_] and 5 equivalents of sodium carbonate made it possible to obtain 20% of the aryldihydronaphthalene derivative and 2% of benzoxanthine derivatives as by-products. At the same time, only 13% of the aryl dihydronaphthalene derivative was isolated when using a similar amount of potassium hexacyanoferrate and 1.5 equivalents of 1% sodium carbonate in chloroform solution at room temperature. Benzoxanthine and naphthalene derivatives emerged as by-products [229] (Figure 6).

Potassium hexacyanoferate was also used in the scheme for the preparation of synthetic lignans to obtain neolignans and further transformation into an aryl dihydronaphthalene product by the Friedel–Craftz reaction in the presence of AlCl_3_ [234]. The resulting product could be further modified in order to synthesize the necessary analogs of natural compounds with biological activity. Figure 6 shows the synthetic pathway for the formation of dimeric products. The oxidative coupling of the protected phenylpropanoid derivative was carried out in a two-phase benzene-water system in the presence of 5.5 equivalents of KOH and 2.5 equivalents of K_3_[Fe(CN)_6_] in an inert atmosphere at room temperature for 30 min. The yield was 92%.

Hydrogen peroxide and horseradish peroxidase (HRP) are also known as catalysts for oxidative dimerization. In 2002, a group of Japanese scientists obtained americanol A and isoamericanol A [235]. Caffeic acid derivatives were used as the starting material, which reacted at room temperature in the presence of H_2_O_2_ in phosphate buffer (0.1 M, pH 6.0) with 18% dioxane and HRP as a catalyst. The main reaction products were benzodioxane derivatives, which were the target natural lignans. Aryl dihydronaphthalene derivatives were found only as by-products in minor amounts (Figure 7). Later publications confirmed the possibilities of HRP use as a catalyst in the presence of hydrogen peroxide for the synthesis of other lignans [236].

Another alternative method of oxidative dimerization is the oxidation of caffeic acid with oxygen at pH~8.5 [237]. The described products obtained by this method were identical to the dimerization method described above using HRP (Figure 7). The exact yield for obtained compounds was not published. However, such method had a rather low selectivity and was not suitable for the targeted preparation of a desired product by oxidative coupling reaction.

The most common method for the synthesis of aryldihydronaphthalene derivatives is oxidative dimerization in the presence of FeCl_3_. In 1989, rabdosiin was obtained for the first time in the form of a racemic mixture [238]. Methyl caffeate was used as the starting material (Figure 8). Next, the oxidative coupling was carried out in acetone at room temperature in the presence of FeCl_3_∙6H_2_O. The yield was 28%. Later, Boguki et al. conducted a study on the production of rabdosiin under varied conditions (changed solvents, the amount of FeCl_3_) [239]. The most optimal conditions for dimerization were a mixture of acetone-water in a ratio of 2:1 and 2.2 equivalents of FeCl_3_. At the same time, the use of THF as a solvent led to decreased selectivity and reaction rate. In the course of the reaction, aryldihydronaphthalene derivatives were formed as a mixture of diastereomers as the main reaction products and benzoxyfuran derivatives as minor products.

In 2021, Belozerova et al. successfully applied the method of oxidative dimerization using FeCl_3_ as an oxidative reagent to produce aryldihydronaphtalene lignan sevanol [121]. Crucial coupling step of isocitrate of caffeic acid was applied using optimal reaction medium carefully selected before [240]. The sevanol molecule was obtained in thirteen synthetic steps from malic acid with a 3% overall yield (Figure 9). The construction of a dihydronaphthalene ring by oxidative dimerization of a protected dihydroxycinnamic acid ester was carefully researched.

### 4.2. Classical Cyclization and Non-Phenolic Oxidative Dimerization

The classical approach to the cyclization of phenylpropanoid subunits by dimerization in the presence of oxidizing agentis often accompanied by low yields due to the formation of a large amount of by-products as discussed above. Undesired by-products formation hampered isolating the target compound from the complex reaction mixture. This problem could be avoided with the use of alternative methods for lignans production via condensation or cyclization. Also, similar methods are used as intermediate stages in order to obtain varied analogues of natural compounds [234,241].

In 2014, Ishikawa et al. showed the complete synthesis of (+)-sesamin using aldol reactions to form the key lactone intermediate [242]. After reduction of lactone and subsequent treatment with HCl in situ, using AcCl in the presence of MeOH, Ishikawa et al. could obtain the desired furofuran ring of (+)-sesamin (Figure 10).

In 1995, Tanaka et al. developed a method for the total synthesis of (+)-schizandrim, (+)-gomisin A and (+)-isoschizandrin, as well as their further optimization, which enabled obtaining similar dibenzoylbutyrolactone lignans with good yield [243]. The demonstrated method includes the use of both oxidative coupling and classical cyclization approaches. The authors obtained a key lactone using aldol condensation, which subsequently underwent oxidative dimerization in the presence of iron (III) perchlorate in trifluoroacetic acid and dichloromethane. Further successive transformations made it possible to obtain (+)-schisandrin, (+)-isoschizandrin and gomisin A (Figure 11).

A similar synthesis strategy allowed Tanaka et al. to obtain other dibenzocyclooctane lignans, namely, gomisin N, J, schizandrin B, (-)-deoxyschizandrin, wuweizisu C, kadsurin (Figure 12) [244].

Fischer et al. demonstrated another example of aldol condensation followed by domino radical reaction in order to build the desired skeleton of lignans which were supposed to be (-)-arctgenin and (-)-matairesinol and its analogues [245]. Thus, they could develop and optimize an elegant approach that helped to produce similar dibenzylbutyrolactones in a good yield (Figure 13).

One of the classical cyclization methods is the Stobbe condensation that is used in the most of cases in order to build the basic skeleton of a desired lignan. In 2010, an efficient route of nordihydroguaiaretic acid synthesis (NGDA), (-)-saururenin and their analogues was described by Xia et al. [246]. The synthetic pathway was based on a unified synthetic strategy involving the Stobbe reaction and subsequent alkylation to construct lignan skeletons. The corresponding aldehyde was chosen as a starting molecules for the condensation reaction with diethyl succinate followed by treatment with LDA and 3,4-methylenedioxybenzyl bromide to give the desired diester as a product of condensation. Next, after the subsequent transformation of key intermediates, it became possible to prepare the target dibenzylbutane lignans (Figure 14).

The Diels-Alder reaction is used to synthesize arylnaphthalene and aryltetralin lignans. This technique included the cyclization of acetylenic anhydride and was used for the synthesis of taiwanin C and dehydrootobain in the 60–70s [247,248]. Figure 15 shows an example of Diels-Alder reaction use to obtain lignans.

In 2019, Chi et al. used Diels-Alder reaction in order to build the key scaffold of podophyllotoxin [249]. Tricyclic Diels-Alder adducts were prepared from cyclobutanol derivative lacking the aryl group in moderate yields. After careful and complicated search, authors could develop an interesting C-H bond arylation strategy that helped to synthesize the target compound podophyllotoxin (Figure 16).

In addition to the strategies mentioned above, a wide range of methods, such as the aldol method of synthesis, rearrangement and cycloaddition, 1,4-addition of an acyl anion to an α-unsaturated carbonyl derivative, photocyclization under the action of UV light are discussed in the literature for lignan synthesis. Each method differs in its key steps and final yield of the target lignan. The chosen dimerization strategy can significantly affect the direction of the reaction in relation to the synthetic production of one or another class of natural lignans.

## 5. Conclusions

Plant lignans are attractive molecules for the drug development. They bear evolutionary optimized pharmacophores and are available for the production by chemical synthesis. In a plethora of studies, lignans have been shown to exert antioxidant, anti-inflammatory, neuroprotective, anti-cancer, and chemopreventive activities. Interestingly, some representatives combine all these activities (arctigenin, NDGA, schisandrin B, sesamin). The antioxidant properties of the most common lignans largely determine their functionality. Thus, the ability of lignans to regulate the ROS/ROS-sensitive proteins ratio makes it possible to switch intracellular signaling pathways. As a rule, antioxidant lignans inhibit ROS-induced activation of the NF-κB pathway, which ultimately leads to a decreased expression of inflammatory cytokines (IL-1β, IL-4, IL-6, TNF-α, MCP-1), TGF-β1 and pro-inflammatory enzymes (COX, LOX, PLA_2_ and PDE). Simultaneous activation of the AMPK and Nrf2 pathways increases the expression of antioxidant-related genes (SOD, GR, GPx, Trx, HO-1, CAT) and promotes secretion of the anti-inflammatory interleukin IL-10. This imbalance in cellular response may be crucial in various pathological conditions, mainly associated with a cell death. In contrast, the imbalance in the intracellular signaling can cause the death of the cells with already impaired signalling, such as cancer cells.

However, an unpleasant fact is the toxic effect of antioxidant lignans on liver and kidneys. These severe side effects can greatly diminish the importance of such compounds as therapeutic agents. The toxicity of most lignans has not been thoroughly studied. One of the most actively and widely studied lignans, NDGA, was banned for topical use in humans due to skin hypersensitivity and was prohibited as a food preservative due to its ability to induce cystic nephropathy in rats [214,250]. Also, chronic administration of arctigenin can lead to significant damage to liver and kidneys [223]. Therefore, the high antioxidant potential of a compound is not equally beneficial to different tissues and organs, and such compounds may impair their vital functions.

More intriguing are the lignans, that have been shown to act directly on specific molecular targets. For example, some lignans with antioxidant properties, such as cobusin and eudesmin, can directly activate CFTR and CaCCgie chloride channels and inhibit ANO1/CaCC channels in addition to radical scavenging activity [251]. Another known lignan sevanol from *Thymus armenicaus* inhibited acid-sensing ion channels ASIC1a and ASIC3 and demonstrated a significant anti-inflammatory effect in rodents [119,120,121]. ASIC channels are known as an important pharmacological target for the treatment of inflammatory and neurodegenerative diseases [252]. Completely diverse compounds isolated from plants are able to modulate these channels [253]. Some of these ligands can act as pH-independent ASIC3 activators [254], yet effectively inhibit ASIC1a channels, and possess anti-inflammatory effect [255,256]. And the fact that sevanol acts on the target that is an ion channel located on the cell surface may explain the relatively low doses (0.1–1 mg/kg) in which this lignan shows anti-inflammatory effects. The example of sevanol shows that lignans can be not only antioxidant compounds, but also rather specific inhibitors of channels important for drug development.

## Data Availability

Not applicable.

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
