# Peer review of "Lignans as Pharmacological Agents in Disorders Related to Oxidative Stress and Inflammation: Chemical Synthesis Approaches and Biological Activities"

_ijms, 2022, doi:10.3390/ijms23116031_

Round 1

Reviewer 1 Report

The manuscript in review is very intersting, but more similar to a book chapter than a scientific article.

Although complete, it is too long and should be shortened or divided in more articles.

An important issue is the lack of information about the data search methodology and sources.

As a consequence, also the related tables are excessive.

For example, Table 1 is too much long, and very difficult to follow. It should subdivided in more tables (o r modules A, B, C,...). In addition, its legend may be easily confused with the main text.

My comment for Tab. 4 is the same as for Tab. 1-

The whole text requires an English language and grammar revision.

Author Response

The manuscript in review is very interesting, but more similar to a book chapter than a scientific article.

In this review manuscript, our goal was to present lignans as a class of plant compounds with a well documented anti-inflammatory and antioxidant activity. We decided to present literature data in the form of detailed tables showing information about the experimental models used, concentrations and dosages of lignans that were applied in vitro, ex vivo and in vivo. The tables make it possible to quickly find all the data available for each lignan and further these experimental results are also discussed in the main body of the manuscript. The main text without large tables has a normal size for a review article, but moving the tables in supplementary can greatly reduce the usefulness of the manuscript as a concise guide to antioxidant and anti-inflammatory lignans. In present time we covered major structural lignans’ subclasses such as arylnaphthalene, aryltetralin, dibenzylbutane, dibenzylbutyrolactol, dibenzylbutyrolactone, furanoid, furofuranoid, and dibenzocyclooctadiene. At the same time, it is worth noting that the manuscript is restricted to “classical lignans” and did not touch upon other undoubtedly interesting groups, such as neolignans, sesquineolignans, dineolignans, norlignans, and hybrid lignans, since this would complicate an already extensive text.

Although complete, it is too long and should be shortened or divided in more articles.

In this review, we have tried to describe lignans have been thoroughly studied and for which antioxidant and anti-inflammatory activity has been demonstrated, We believe that it will be difficult to divide the manuscript into several articles, since the structural and synthetic parts are common. However, we will not object to splitting the manuscript into several parts by editors.

An important issue is the lack of information about the data search methodology and sources.

You are absolutely right it is potent to know the freshness of material reviewed. But the lack of a section of materials and methods led to the omission of methodology details of data collection. We retrieved literature available in PubMed, Scopus and Web of Science databases from 2000 to 2022 using queries: “lignans and antioxidants”, “lignans and oxidative stress”, “lignans and free radical scavenging”, “lignans and anti-inflammatory”, “lignans and inflammation”. The active compounds were further divided into eight subgroups of lignans that exhibit antioxidant and/or anti-inflammatory effects (some insufficiently validated compounds were rejected). In the second round of data collection, the same queries were used as in the first round, but the keyword "lignans" was changed to the name of the specific active lignan. We will not include details of search in revised article except the range of publication, since manuscript is already large.

As a consequence, also the related tables are excessive.

We do not agree with this comment, since as it was mentioned above, all tables concentrate potent information for each lignan, which allow to compare results from different papers and go to the should reference.

For example, Table 1 is too much long, and very difficult to follow. It should subdivided in more tables (o r modules A, B, C,...). In addition, its legend may be easily confused with the main text. My comment for Tab. 4 is the same as for Tab. 1-

Since the names of experimental models contain a large number of abbreviations that are of great importance for the perception of data, we considered it necessary to fully decipher these abbreviations in the legends to the tables. If we showed abbreviations in tables, it would complicate them.

The whole text requires an English language and grammar revision.

We subjected the manuscript to language and grammar revision.

Reviewer 2 Report

The review entitled "Lignans as pharmacological agents in disorders related to oxidative stress and inflammation: chemical synthesis approaches and biological activities" regards plant lignans as bioactive agents in disorders associated with oxidative stress and inflammation.

The review is well written and organized although the methodology of the bibliographic research is not described. The authors should clearly define which keywords have been used to search the articles related to the topic. The years considered should also defined. The authors selected only the lignans represented in fig. 1 but several lignans are present in the plant kingdom and new ones have been discovered in the years and their antioxidant and anti-inflammatory activities have been studied.  The authors should justify their choice.

Considering the years 2020 and 2021 the following articles were not considered:

1) Zhang, Y., Lei, Y., Yao, X., Yi, J., Feng, G. Pinoresinol diglucoside alleviates ischemia/reperfusion-induced brain injury by modulating neuroinflammation and oxidative stress(2021) Chemical Biology and Drug Design, 98 (6), pp. 986-996.

2) Yang, Y., Jian, Y., Cheng, S., Jia, Y., Liu, Y., Yu, H., Cao, L., Li, B., Peng, C., Iqbal Choudhary, M., Rahman, A.-U., Wang, W. Dibenzocyclooctadiene lignans from Kadsura coccinea alleviate APAP-induced hepatotoxicity via oxidative stress inhibition and activating the Nrf2 pathway in vitro

3) Tao, W., Hu, Y., Chen, Z., Dai, Y., Hu, Y., Qi, M. Magnolol attenuates depressive-like behaviors by polarizing microglia towards the M2 phenotype through the regulation of Nrf2/HO-1/NLRP3 signaling pathway(2021) Phytomedicine, 91, art. no. 153692, .

4) Kartha, S., Weisshaar, C.L., Pietrofesa, R.A., Christofidou-Solomidou, M., Winkelstein, B.A. Synthetic secoisolariciresinol diglucoside attenuates established pain, oxidative stress and neuroinflammation in a rodent model of painful radiculopathy (2020) Antioxidants, 9 (12), art. no. 1209, pp. 1-16.

5) Ye, H., Sun, L., Li, J., Wang, Y., Bai, J., Wu, L., Han, Q., Yang, Z., Li, L. Sesamin attenuates carrageenan-induced lung inflammation through upregulation of A20 and TAX1BP1 in rats (2020) International Immunopharmacology, 88, art. no. 107009, .

6) Cui, L., Zhu, W., Yang, Z., Song, X., Xu, C., Cui, Z., Xiang, L. Evidence of anti-inflammatory activity of Schizandrin A in animal models of acute inflammation (2020) Naunyn-Schmiedeberg's Archives of Pharmacology, 393 (11), pp. 2221-2229.

7) Li, K., Lv, C. Intradiscal injection of sesamin protects from lesion-induced degeneration (2020) Connective Tissue Research, 61 (6), pp. 594-603.

8) Yoon, J.J., Lee, H.K., Kim, H.Y., Han, B.H., Lee, H.S., Lee, Y.J., Kang, D.G. Sauchinone protects renal mesangial cell dysfunction against angiotensin ii by improving renal fibrosis and inflammation (2020) International Journal of Molecular Sciences, 21 (19), art. no. 7003, pp. 1-14.

9) Yang, S.-R., Hsu, W.-H., Wu, C.-Y., Shang, H.-S., Liu, F.-C., Chen, A., Hua, K.-F., Ka, S.-M. Accelerated, severe lupus nephritis benefits from treatment with honokiol by immunoregulation and differentially regulating NF-κB/NLRP3 inflammasome and sirtuin 1/autophagy axis (2020) FASEB Journal, 34 (10), pp. 13284-13299.

10) Gui, Y., Yang, Y., Xu, D., Tao, S., Li, J. Schisantherin A attenuates sepsis-induced acute kidney injury by suppressing inflammation via regulating the NRF2 pathway (2020) Life Sciences, 258, art. no. 118161, .

11) Kumar, S.S., Hira, K., Begum Ahil, S., Kulkarni, O.P., Araya, H., Fujimoto, Y. New synthetic coumarinolignans as attenuators of pro-inflammatory cytokines in LPS-induced sepsis and carrageenan-induced paw oedema models (2020) Inflammopharmacology, 28 (5), pp. 1365-1373.

12) Zhao, B., Xia, B., Li, X., Zhang, L., Liu, X., Shi, R., Kou, R., Liu, Z., Liu, X. Sesamol supplementation attenuates DSS-induced colitis via mediating gut barrier integrity, inflammatory responses, and reshaping gut microbiome (2020) Journal of Agricultural and Food Chemistry, 68 (39), pp. 10697-10708.

13) Zeng, M., Li, M., Chen, Y., Zhang, J., Cao, Y., Zhang, B., Feng, W., Zheng, X., Yu, Z. A new bisepoxylignan dendranlignan A isolated from Chrysanthemum Flower inhibits the production of inflammatory mediators via the TLR4 pathway in LPS-induced H9c2 cardiomyocytes (2020) Archives of Biochemistry and Biophysics, 690, art. no. 108506, .

The articles above are not exaustive of the database research, more articles could be missed. I suggest to revise the methodology on the database research or if the above articles were out of the scope of the present review the authors should better specify which articles have been considered.

Author Response

The review entitled "Lignans as pharmacological agents in disorders related to oxidative stress and inflammation: chemical synthesis approaches and biological activities" regards plant lignans as bioactive agents in disorders associated with oxidative stress and inflammation.

The review is well written and organized although the methodology of the bibliographic research is not described. The authors should clearly define which keywords have been used to search the articles related to the topic. The years considered should also defined.

You are absolutely right it is potent to know the freshness of material reviewed. But the lack of a section of materials and methods led to the omission of methodology details of data collection. We retrieved literature available in PubMed, Scopus and Web of Science databases from 2000 to 2022 using queries: “lignans and antioxidants”, “lignans and oxidative stress”, “lignans and free radical scavenging”, “lignans and anti-inflammatory”, “lignans and inflammation”. The active compounds were further divided into eight subgroups of lignans that exhibit antioxidant and/or anti-inflammatory effects (some insufficiently validated compounds were rejected). In the second round of data collection, the same queries were used as in the first round, but the keyword "lignans" was changed to the name of the specific active lignan. We will not include details of search in revised article except the range of publication, since manuscript is already large.

The authors selected only the lignans represented in fig. 1 but several lignans are present in the plant kingdom and new ones have been discovered in the years and their antioxidant and anti-inflammatory activities have been studied. The authors should justify their choice.

In this review, we tried to present lignans as a wide class of plant compounds with an anti-inflammatory and antioxidant activity. The size limitation led to the selection of only "classical lignans", such as arylnaphthalene, aryltetralin, dibenzylbutane, dibenzylbutyrolactol, dibenzylbutyrolactone, furanoid, furofuranoid, and dibenzocyclooctadiene lignans as an object for review, and did not allow other interesting groups, such as neolignans, sesquineolignans, dineolignans, norlignans and hybrid lignans, to be reviewed. Among “classical lignans”, the one that had both antioxidant and anti-inflammatory activity was preferably chosen. To reduce manuscript size only the compound that was more completely and qualitatively characterized in experiments in vitro and in vivo, was subjected to the review.

Considering the years 2020 and 2021 the following articles were not considered:

1) Zhang, Y., Lei, Y., Yao, X., Yi, J., Feng, G. Pinoresinol diglucoside alleviates ischemia/reperfusion-induced brain injury by modulating neuroinflammation and oxidative stress(2021) Chemical Biology and Drug Design, 98 (6), pp. 986-996.

This article has already been reviewed in this manuscript ( new ref. 156).

2) Yang, Y., Jian, Y., Cheng, S., Jia, Y., Liu, Y., Yu, H., Cao, L., Li, B., Peng, C., Iqbal Choudhary, M., Rahman, A.-U., Wang, W. Dibenzocyclooctadiene lignans from Kadsura coccinea alleviate APAP-induced hepatotoxicity via oxidative stress inhibition and activating the Nrf2 pathway in vitro

We excluded this article because authors attempted to demonstrate the antioxidant effect of heilaohusuin B in the APAP-induced cytotoxicity test in just one in vitro experiment. However, our confusion was that APAP itself caused a significant increase of Nrf2 and HO-1 concentration, and the addition of heilaohusuin B only enhanced this effect. Therefore, by our opinion, it is not correct to correlate the hepatoprotective effect of heilaohusuin B with Nrf2 pathway activation.

3) Tao, W., Hu, Y., Chen, Z., Dai, Y., Hu, Y., Qi, M. Magnolol attenuates depressive-like behaviors by polarizing microglia towards the M2 phenotype through the regulation of Nrf2/HO-1/NLRP3 signaling pathway(2021) Phytomedicine, 91, art. no. 153692.

As we indicated in the Introduction, this review is limited to a specific structural group of lignans. Since magnolol belongs to the neolignan group, we did not include it.

4) Kartha, S., Weisshaar, C.L., Pietrofesa, R.A., Christofidou-Solomidou, M., Winkelstein, B.A. Synthetic secoisolariciresinol diglucoside attenuates established pain, oxidative stress and neuroinflammation in a rodent model of painful radiculopathy (2020) Antioxidants, 9 (12), art. no. 1209, pp. 1-16.

This article has already been reviewed in this manuscript (new ref. 143).

5) Ye, H., Sun, L., Li, J., Wang, Y., Bai, J., Wu, L., Han, Q., Yang, Z., Li, L. Sesamin attenuates carrageenan-induced lung inflammation through upregulation of A20 and TAX1BP1 in rats (2020) International Immunopharmacology, 88, art. no. 107009.

This article data were added in the manuscript (ref. 205).

6) Cui, L., Zhu, W., Yang, Z., Song, X., Xu, C., Cui, Z., Xiang, L. Evidence of anti-inflammatory activity of Schizandrin A in animal models of acute inflammation (2020) Naunyn-Schmiedeberg's Archives of Pharmacology, 393 (11), pp. 2221-2229.

This article data were added in the manuscript (ref. 174).

7) Li, K., Lv, C. Intradiscal injection of sesamin protects from lesion-induced degeneration (2020) Connective Tissue Research, 61 (6), pp. 594-603.

Since the sesamine properties are shown in full in this review, and its various antioxidant and anti-inflammatory effects are also presented in many models, we will allow ourselves to dismiss this article that is not directly related to the topic.

8) Yoon, J.J., Lee, H.K., Kim, H.Y., Han, B.H., Lee, H.S., Lee, Y.J., Kang, D.G. Sauchinone protects renal mesangial cell dysfunction against angiotensin ii by improving renal fibrosis and inflammation (2020) International Journal of Molecular Sciences, 21 (19), art. no. 7003, pp. 1-14.

This article data were added in the manuscript (ref. 35).

9) Yang, S.-R., Hsu, W.-H., Wu, C.-Y., Shang, H.-S., Liu, F.-C., Chen, A., Hua, K.-F., Ka, S.-M. Accelerated, severe lupus nephritis benefits from treatment with honokiol by immunoregulation and differentially regulating NF-κB/NLRP3 inflammasome and sirtuin 1/autophagy axis (2020) FASEB Journal, 34 (10), pp. 13284-13299.

Since honokilol also (like magnolol) belongs to neolignans, we did not take it into consideration.

10) Gui, Y., Yang, Y., Xu, D., Tao, S., Li, J. Schisantherin A attenuates sepsis-induced acute kidney injury by suppressing inflammation via regulating the NRF2 pathway (2020) Life Sciences, 258, art. no. 118161.

This article data were added in the manuscript (ref. 55).

11) Kumar, S.S., Hira, K., Begum Ahil, S., Kulkarni, O.P., Araya, H., Fujimoto, Y. New synthetic coumarinolignans as attenuators of pro-inflammatory cytokines in LPS-induced sepsis and carrageenan-induced paw oedema models (2020) Inflammopharmacology, 28 (5), pp. 1365-1373.

Coumarinolignans belong to a group of non-conventional lignans similar to favonolignans and stilbenolignans and were not considered in this review.

12) Zhao, B., Xia, B., Li, X., Zhang, L., Liu, X., Shi, R., Kou, R., Liu, Z., Liu, X. Sesamol supplementation attenuates DSS-induced colitis via mediating gut barrier integrity, inflammatory responses, and reshaping gut microbiome (2020) Journal of Agricultural and Food Chemistry, 68 (39), pp. 10697-10708.

Sesamol belongs to a simple phenols (benzodioxolol) structural group and also was not considered in this review.

13) Zeng, M., Li, M., Chen, Y., Zhang, J., Cao, Y., Zhang, B., Feng, W., Zheng, X., Yu, Z. A new bisepoxylignan dendranlignan A isolated from Chrysanthemum Flower inhibits the production of inflammatory mediators via the TLR4 pathway in LPS-induced H9c2 cardiomyocytes (2020) Archives of Biochemistry and Biophysics, 690, art. no. 108506.

This article data were added in the manuscript (ref. 103).

The articles above are not exhaustive of the database research, more articles could be missed. I suggest to revise the methodology on the database research or if the above articles were out of the scope of the present review the authors should better specify which articles have been considered.

Based on the reviewer suggestion and utilizing mentioned above criteria we reanalyze data collected and included additional articles in the revised version (see ref. Wang et al., 2015; Desai et al., 2014; During et al., 2012; Spilioti et al., 2014; Laavola et al., 2017; Nakano et al., 2017). Taking into account reviewer 1 comment about the large size of the manuscript, we did not include all available information about lignans, but limited ourselves to the well studied compounds and include representatives for maximally possible number of structural subgroups.

Round 2

Reviewer 2 Report

The following reviews published in "International Journal of Molecular Sciences" have a section describing the methodology used to find the articles related to the topic of the present review.

"Proteomic Analysis of Human Sputum for the Diagnosis of Lung Disorders: Where Are We Today?" Int. J. Mol. Sci. 2022, 23, 5692. https://doi.org/10.3390/ijms23105692

"Fascial Innervation: A Systematic Review of the Literature" Int. J. Mol. Sci. 2022, 23, 5674. https://doi.org/10.3390/ijms23105674

"Probiotics Function in Preventing Atopic Dermatitis in Children" Int. J. Mol. Sci. 2022, 23, 5409. https://doi.org/10.3390/ijms23105409

Since the present review is very wide and many articles were missed, it is important for readers to know which databases were used and which keywords have been used.